# Structural basis of double-stranded RNA recognition by the J2 monoclonal antibody

Charles Bou-Nader [1,5], Kevin M. Juma[1,5], Ankur Bothra [2,5], Andrew J. Brasington[1], Rodolfo Ghirlando[1], Motoshi Suzuki [3], David N. Garboczi[4], Stephen H. Leppla[2] ✉ & Jinwei Zhang [1] ✉

Double-stranded (ds) RNAs are major structural components of the transcriptome, hallmarks of viral infection, and primary triggers of innate immune responses. The J2 monoclonal antibody is the gold-standard method to discover and map endogenous dsRNAs across subcellular locations and cell surfaces, detect exogenous RNAs in viral infection, and surveil mRNA prophylactics and therapeutics for inflammatory dsRNAs. To define its epitope, specificity, and mechanism, we determine a 2.85 Å co-crystal structure of J2 antigen-binding fragment (Fab) bound to dsRNA. J2 uses its heavy and light chains in tandem to track the dsRNA minor groove, recognizing a staggered 8-bp duplex. J2 is highly selective for dsRNAs, requires 14 bp for robust binding, and exhibits greatly diminished binding for GC-rich dsRNAs. J2 and the R-loop-specific S9.6 antibody share a common recognition strategy distinct from intracellular dsRNA-binding proteins. This study provides mechanistic insights into dsRNA recognition and establishes a framework for reliable application and data interpretation of the J2 antibody in RNA discovery.

Double-stranded RNA (dsRNA) is a major structural form of cellular coding and noncoding RNAs. Rigid, persistent dsRNA segments assemble into tRNAs, ribosomal RNAs, catalytic and regulatory RNAs including ribozymes and riboswitches, mRNAs (particularly those with long, structured UTRs), long noncoding RNAs (lncRNAs), transcripts from LINE and SINE retrotransposons including Alu elements, and more[1–3]. Nuclear export of mRNAs and lncRNAs was recently suggested to be facilitated by transient dsRNA formation with their antisense transcripts[4], as dsRNAs are preferentially loaded onto RNA export receptors such as Mex67. Abundant dsRNAs are found in the nucleus, cytoplasm, and organelles such as mitochondria due to bidirectional transcription. Excitingly, dsRNAs are increasingly discovered to populate cellular surfaces (e.g., glycoRNAs), extracellular space, including biofilm, tissues and biological fluids[5–8].

DsRNA is also a prominent pathogen-associated molecular pattern (PAMP) that frequently accumulates as a result of viral infection and replication[1]. DsRNAs constitute the genomes of dsRNA viruses such as the rotavirus—the leading cause of acute gastroenteritis in infants and children, the ruminant-infecting Bluetongue virus (BTV), plant-infecting Rice dwarf virus (RDV), and *Pseudomonas*-infecting Bacteriophage Φ6. DsRNAs are also major structural components of the RNA genomes of single-stranded (ss) RNA viruses and retroviruses such as HIV-1. Besides serving as viral genomes, dsRNAs are frequently produced as replication intermediates of dsDNA, ssDNA, and ssRNA viruses or from bidirectional transcription of viral genomes. As a result, dsRNA has emerged as the principal trigger of innate antiviral immune responses in metazoans and is subject to direct detection by a cohort of pattern recognition receptors (PRRs), including RIG-I, MDA5, LGP2, TLRs, OASs, and PKR[1,9]. Activation of such dsRNA receptors produces

[1]Laboratory of Molecular Biology, National Institute of Diabetes and Digestive and Kidney Diseases, Bethesda, MD, USA. [2]Laboratory of Parasitic Diseases, Division of Intramural Research, National Institute of Allergy and Infectious Diseases, Bethesda, MD, USA. [3]Protein & Chemistry Section, Research Technologies Branch, National Institute of Allergy and Infectious Diseases, Rockville, MD, USA. [4]Structural Biology Section, Research Technologies Branch, National Institute of Allergy and Infectious Diseases, Bethesda, MD, USA. [5]These authors contributed equally: Charles Bou-Nader, Kevin M. Juma, Ankur Bothra. ✉e-mail: sleppla@niaid.nih.gov; jinwei.zhang@nih.gov

type I interferons and other proinflammatory cytokines, represses global translation and reprograms cellular metabolism towards stress response and apoptosis. Due to the immunogenic and inflammatory properties of dsRNAs, they are either prevented from significant accumulation by RNases, edited by adenosine deaminases acting on RNA (ADARs) to reduce their base-pairing and dsRNA characteristics, or sequestered away from the PRRs through protein binding or sub-cellular compartmentalization in normal nucleic acid metabolism. Dysregulation of dsRNA metabolism is a significant contributor to inflammatory and autoimmune diseases including systemic lupus erythematosus (SLE), inflammatory bowel disease (IBD), type 1 diabetes, and psoriasis[10–13].

The J2 monoclonal antibody is the principal tool to map and track endogenous dsRNAs, detect foreign dsRNAs in viral infection, discover and characterize new viruses and RNAs, and surveil mRNA prophylactics and therapeutics for dsRNA contamination[14–18]. Despite its widespread use, it remains unknown how the J2 antibody recognizes dsRNA and its selectivity towards different types, lengths, and sequences of nucleic acids. Based on an atomic force microscopy (AFM) analysis, it was estimated that J2 requires ~40 bp (base-pair) dsRNA for binding. J2 exhibits a possible preference for A-rich sequences and binds inefficiently to poly(rI)·poly(rC)[15,18]. The indeterminate J2 specificity, undefined epitope on the dsRNA, potential sequence bias, and unknown mechanism of action have presented significant challenges in its numerous applications. For instance, it has confounded the determination of whether negative-strand RNA viruses produce substantial quantities of dsRNAs in their life cycle[19].

In this work, to address these questions and enable, optimize, and delimit reliable application of and data interpretation for the J2 antibody, we conduct biophysical analyses to define its selectivity, specificity, and binding kinetics. We determined a 2.85 Å co-crystal structure of the J2 Fab bound to a 23-bp dsRNA, which reveals its epitope and unusual mechanism of recognition. We further probe the observed J2-dsRNA interface using mutational analyses and examine the effects of nucleic acid type, length, and sequence on J2 recognition. Finally, we compare the dsRNA recognition strategies employed by secreted antibodies *versus* those by intracellular proteins.

## Results

### J2 antibody exhibits high specificity and complex binding kinetics for dsRNAs

Using the antibody-expression platform we recently developed for the S9.6 antibody[20,21], we produced both J2 IgG and Fab and performed initial dsRNA binding characterizations using size-exclusion chromatography with multi-angle light scattering (SEC-MALS) and sedimentation velocity-analytical ultracentrifugation (SV-AUC). SEC-MALS revealed that the J2 Fab bound a 27-bp dsRNA forming a 1:1 complex (Fig. 1a, b, Supplementary Fig. 1). AUC analysis further detected stable 1:1 complexes that formed between the J2 Fab and a 23-bp dsRNA, and between the J2 IgG and a 30-bp dsRNA (Supplementary Fig. 1). These analyses confirmed the functional competency of the J2 IgG and Fab we produced and their stoichiometric interactions with dsRNAs of varying lengths.

To define the nucleic acid selectivity of the J2 antibody, we employed biolayer interferometry (BLI) to evaluate its interaction with dsRNA, dsDNA, ssRNA, ssDNA, and RNA-DNA hybrid of the same length (40 nucleotides (nts) per strand) and sequence (Fig. 1a). J2 IgG was immobilized on Protein-A probes and served as the ligand, whereas different nucleic acids were used as analytes. While robust binding by dsRNA was detected, dsDNA, ssRNA, and ssDNA exhibited no appreciable binding to J2 under the same conditions (Fig. 1c, d). Remarkably, J2 robustly discriminates against RNA-DNA hybrids, which form A-form duplex structures similar to dsRNA. This ensures that J2 does not substantially recognize abundant R-loops, which cover 5–10% of bacterial, eukaryotic, and viral genomes[22]. By contrast, the R-loop-

**Table 1 | Summary of X-ray crystallographic data collection and refinement statistics**

|  | J2 Fab bound to dsRNA |
| --- | --- |
| **Data collection** |  |
| Space group | $P\,2\,2_1\,2_1$ |
| Cell dimensions |  |
| $a, b, c$ (Å) | 65.391 92.374 194.663 |
| $\alpha, \beta, \gamma$ (°) | 90, 90, 90 |
| Resolution (Å) | 54.28–2.85 (2.90–2.85) |
| $R_{sym}$ or $R_{merge}$ | 0.452 (2.77) |
| $R_{pim}$ | 0.085 (0.54) |
| $I\,/\,\sigma I$ | 7.3 (1.0) |
| $CC_{1/2}$ | 0.992 (0.444) |
| Completeness (%) | 100.0 (98.1) |
| Redundancy | 29.2 (26.6) |
| **Refinement** |  |
| Resolution (Å) | 20.64–2.85 (2.96–2.85) |
| No. reflections | 28092 (3013) |
| $R_{work}\,/\,R_{free}$ | 0.212/0.262 (0.297/0.369) |
| No. atoms | 7309 |
| Macromolecule | 7234 |
| Water | 26 |
| Ligand | 49 |
| $B$-factors | 50.82 |
| Macromolecule | 50.84 |
| Water | 36.61 |
| Ligands | 55.69 |
| R.m.s. deviations |  |
| Bond lengths (Å) | 0.002 |
| Bond angles (°) | 0.52 |
| Maximum-likelihood coordinate precision (Å) | 0.36 |
| Protein Data Bank (PDB) accession code | 9OJV |

specific S9.6 antibody exhibits substantial cross-reactivity with dsRNAs[20,23].

Kinetic analyses of dsRNA binding to J2 IgG consistently produced biphasic sensorgrams (Fig. 1c, Supplementary Figs. 2–4 and Table 1), despite their stoichiometric (1:1) binding observed by AUC (Supplementary Fig. 1). The BLI sensorgrams deviate substantially from a simple, homogeneous binding model and require a heterogeneous ligand model which produced excellent agreement (Supplementary Fig. 3). This suggests the presence of two types of J2 ligands on the surface, or two separable binding modes. To ensure that non-specific surface interactions did not produce the ligand heterogeneity, we first verified the complete absence of RNA binding without J2. We then examined dsRNA binding across a wide range of ionic strengths, orbital speeds, and dsRNA lengths (Fig. 1e–j, Supplementary Figs. 2–4). These additional analyses suggest that dsRNA association with J2 IgG is an inherently heterogeneous process across multiple solute conditions, dsRNA lengths and sequences. By contrast, an Adenovirus Virus-Associated RNA I (VA–I, Supplementary Fig. 2i) bound J2 IgG following a homogeneous binding model, under the same conditions and across several ionic strengths (Fig. 1c, Supplementary Figs. 2, 3). VA-I is a prominent viral mimic of dsRNA that inhibits antiviral Protein Kinase R (PKR) by entrapping PKR monomers using its coaxially stacked apical stem and tetrastem, thereby preventing PKR dimerization and activation[9,24]. Notably, VA-I dissociates from J2 much more slowly than dsRNA, a characteristic that befits its role in recruiting and sequestering the dsRNA-binding motifs (dsRBMs) of PKR (Supplementary Fig. 3b). In addition, VA-I binding to J2 exhibits higher sensitivity to

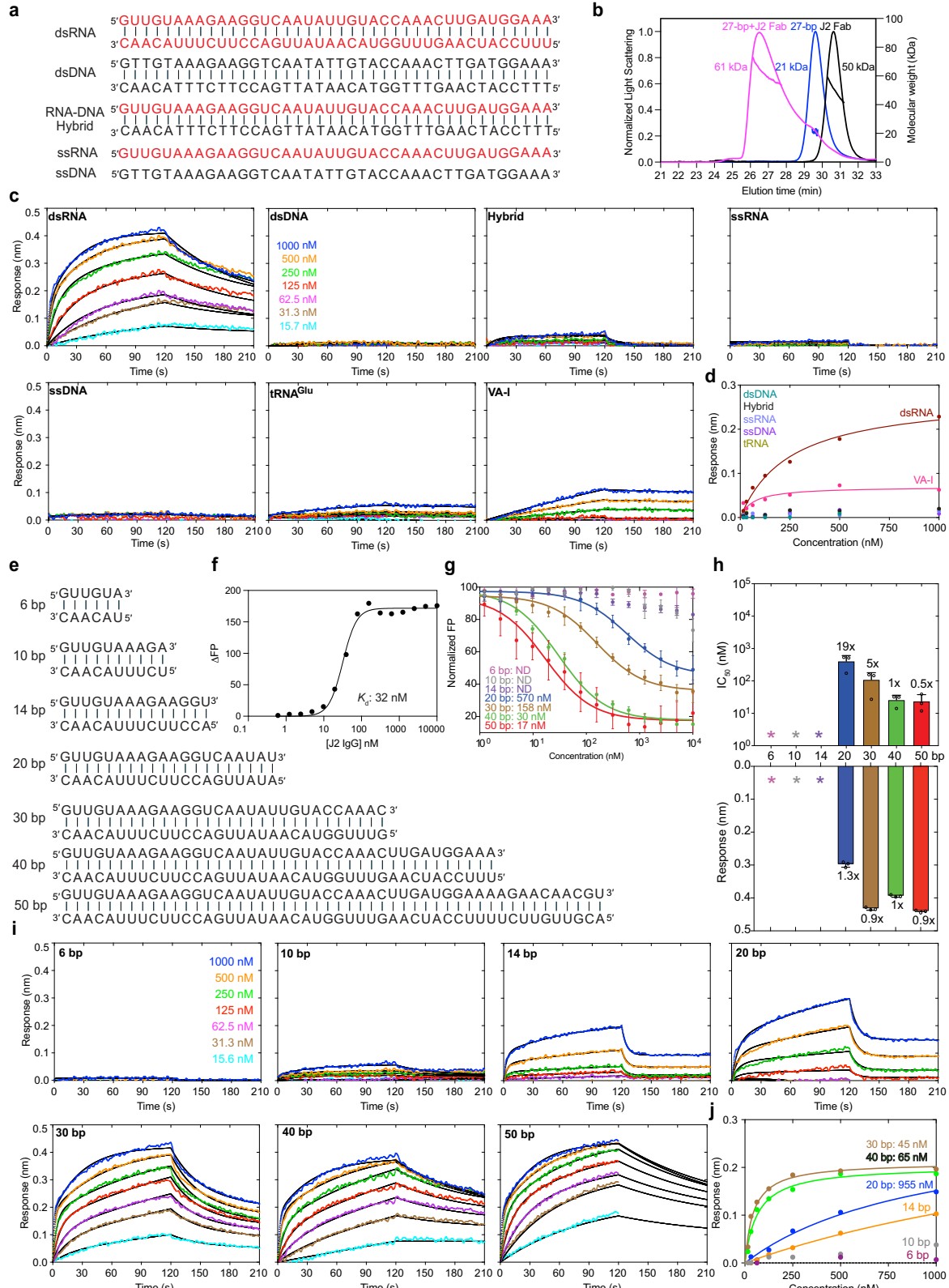

ionic strength than dsRNAs, which could be due to additional, sequence-specific electrostatic contacts that VA–I was proposed to make (Supplementary Fig. 3b, c)[9,24]. The distinct behaviors of dsRNA and VA–I suggest that dsRNA elements of different sequences and structures can differentially interact with J2. The observed heterogeneous binding behavior of dsRNA to J2 IgG is likely due to the presence of multiple, non-identical epitopes along the dsRNA, or J2 sliding

along the dsRNA, as reported for dsRBMs[25]. It could also stem from differences in the number of Fabs contacting the dsRNA, paratope conformations or accessibility between the two Fabs due to asymmetric surface immobilization, or the location, orientation, or conformation of the dsRNA epitope during the initial encounter.

The remarkable selectivity of the J2 antibody for dsRNA over hybrids, dsDNA, ssRNA, and ssDNA gives essential assurances that

**Fig. 1 | Nucleic acid specificity and dsRNA length requirements of J2 antibody.**
**a** Sequences of single-stranded (ss) and double-stranded (ds) nucleic acids used in J2 binding analyses. RNA strands are shown in red; DNA in black. **b** SEC-MALS analysis of J2 Fab binding to a 27-bp dsRNA. **c** BLI sensorgrams using J2 IgG as immobilized ligand and the nucleic acids in (**a**) as analytes, at 400 mM KCl. **d** Plot of sensor responses (nm) against nucleic acid concentrations in BLI steady-state analysis. **e** dsRNAs of various lengths used for J2 binding analyses. **f** Bimolecular fluorescence polarization (FP) analysis of J2 IgG binding to a 3′-FAM-labeled Adenovirus VA-I RNA. ΔFP: change in FP expressed in dimensionless millipolarization

(mP) units. Experiments were performed twice. **g** Competition FP analysis of dsRNAs in (**e**) in displacing 3′-FAM labeled VA-I RNA prebound to J2 IgG. Apparent $IC_{50}$s are indicated. **h** $IC_{50}$s derived from FP in (**g**) (upper) and maximum BLI sensor responses from (**i**) (lower). **i** BLI sensorgrams of dsRNAs in (**e**) binding to immobilized J2 IgG. Due to significant dependency of J2 binding on ionic strengths, 100 mM KCl was used for 6, 10, 14, and 20 bp; 200 mM KCl for 30 bp; 400 mM KCl for 40 and 50 bp dsRNA. **j** Steady-state analyses of BLI sensorgrams in (**i**). Apparent $K_d$s are indicated. Values are mean ± s.d. of three biologically independent samples unless otherwise indicated.

previous and future studies using the J2 antibody provide reliable detection of dsRNAs among numerous competing nucleic acids. Further, the drastic differences between perfectly paired dsRNAs and natural RNAs such as VA-I in J2 binding kinetics and stability suggest J2 does not uniformly detect all dsRNAs in cells, indicating a selectivity for dsRNA length, sequence, or structure.

## J2 IgG binds dsRNA in a length-dependent manner

As a widely used tool to detect, image, and enrich dsRNAs of cellular and viral origins, it is important to accurately define the effective dsRNA length required for J2 recognition. A previous AFM analysis estimated a 40-bp length requirement for J2 binding[15]. However, we observed robust and stable binding of J2 to VA-I RNA, which has a maximal dsRNA length of ~27 bp (Fig. 1f, Supplementary Figs. 2, 3). To quantitatively assess the effect of dsRNA length on J2 binding, we employed a fluorescence polarization (FP) titration assay. First, we measured direct, bimolecular binding of J2 IgG to a 3′-FAM (fluorescein)-labeled VA-I RNA, which yielded an apparent $K_d$ of ~32 nM (Fig. 1f, Supplementary Fig. 3d). Then, we evaluated the ability of dsRNAs of varying lengths to displace the VA-I RNA pre-bound to J2 IgG. While a 6-bp dsRNA could not compete with VA−I at all, the 10-bp and 14-bp dsRNAs exhibited weak but detectable binding to J2 (Fig. 1g). Further elongating the dsRNA progressively and substantially improved binding, attaining an $IC_{50}$ of ~17 nM at 50 bp (Fig. 1g, h).

To corroborate and extend the FP findings using an orthogonal method, we employed additional BLI analyses. First, we observed that J2 binding to dsRNA is sensitive to the ionic strength, suggesting contributions from electrostatic interactions at the interface (Supplementary Fig. 3). At a near-physiological 100 mM KCl, the 6-bp dsRNA produced no response on the BLI sensor (Fig. 1i, j, Supplementary Fig. 4). The 10-bp dsRNA started to trigger a weak, dose-dependent binding signal. At 14-bp and 20-bp, the dsRNAs yielded robust association responses but also dissociated rapidly from the J2 IgG (Fig. 1i, j, Supplementary Fig. 4 and Table 1). The association signal approached a plateau at 30 and 40 bp dsRNA lengths, while the dissociation further slowed with increasing dsRNA lengths (Fig. 1h–j, Supplementary Fig. 4). At 50-bp length, the dissociation portion of the sensorgrams could not be reliably fit by the heterogeneous ligand model, likely due to local rebinding of longer dsRNAs, or dsRNA being sufficiently long to bridge adjacent IgGs. Overall, the congruent BLI and FP data suggest a minimal length threshold of ~14 bp dsRNA for practical detection by the J2 antibody, and near-optimal J2 binding and retention at dsRNA lengths longer than 30 bp. This length dependency rationalizes our observations that J2 does not substantially bind tRNAs but strongly associates with VA−I, which possess contiguous dsRNA segments of 10–12 bp and 27 bp, respectively (Supplementary Fig. 2h, i).

## Co-crystal structure of a J2 Fab bound to 23-bp dsRNA

To visualize how the J2 antibody specifically recognizes dsRNA, discern its minimal epitope, and understand its observed binding preferences, we co-crystallized a J2 Fab bound to a 23-bp dsRNA (Fig. 2a, Table 1 and Supplementary Fig. 5). The structure captured two J2 Fabs bound to the opposite sides of a central, 17-bp portion of the same dsRNA *in crystallo*, via nearly identical interfaces (all-atom RMSD ~ 0.6 Å, Fig. 2a, b, d, Supplementary Figs. 5, 6). While the central 17-bp dsRNA segment

exhibited well-defined electron density, both flanking 3-bp terminal regions had only weak or no density. This finding is consistent with the notion that terminal regions of dsRNAs, such as HIV-1 TAR, exhibit substantial conformational flexibility and engage in transient excursions[26,27]. Each J2 Fab footprint on the dsRNA spans approximately 8 bp (Fig. 2b, c, f), which rationalizes the complete lack of binding by the 6-bp dsRNA and emergence of weak binding at 10 bp (Fig. 1i). The structure reveals that J2 uses its heavy chain (HC) and light chain (LC) complementarity-determining regions (CDRs) in a tandem arrangement to continuously track a single minor groove of the dsRNA, similar to how the S9.6 antibody follows the minor groove of RNA-DNA hybrid duplexes[20]. J2 binding does not substantially deform the dsRNA, reducing the entropic cost of complex formation (Fig. 2e).

Interestingly, the J2 Fab asymmetrically binds the two RNA strands that bookend the minor groove in a staggered configuration (Fig. 2c, f). While it makes contacts to seven consecutive nucleotides on one strand (C15-U21 of Chain E), it touches six consecutive nucleotides on the opposite strand (U6-A11 of Chain F), with a 2-bp stagger or offset. As a result, an 8-bp dsRNA constitutes the minimal physical epitope required for J2 binding. Recognition involves a combination of 2′-OH groups, nucleobase edges, and backbone phosphate oxygens (Figs. 2f, 3a, b). Like the S9.6 antibody, the J2 heavy chain CDRs primarily mediate duplex recognition, while the light chain CDRs play assistive roles by positioning and stabilizing the heavy chain CDRs (Fig. 3a, b). Among the three heavy chain CDRs, CDR-H1 (denoting the first CDR of the heavy chain variable region) makes a relatively minor contribution, while CDR-H2 and CDR-H3 play essential roles in binding dsRNA (Figs. 2c, f, 3a, b). CDR-H1 contains a short $3_{10}$ helix, which positions the peptide backbone carbonyl between N31 and H32 to contact the 2′-OH of A17, near the center of the RNA epitope (Fig. 3a, b). Since the side chains of this $3_{10}$ helix do not directly bind the RNA, we instead asked if disrupting its main chain helical geometry by an A30P substitution would impact dsRNA binding. Surprisingly, the A30P substitution in the $3_{10}$ helix is well tolerated, suggesting that the helical conformation is not required to engage the RNA (Fig. 3c, d, Supplementary Fig. 7). It is possible that the increased hydrophobic character of proline helped stabilize the local structure without the $3_{10}$ helix.

On one flank of the centrally located CDR-H1, the short CDR-H2 binds at the edge of the epitope across the minor groove. It presents a network of three closely spaced tyrosines, Y50, Y52, and Y54, which collectively recognize the U9·A16 base pair. Specifically, Y50 binds the 2′-OH and phosphate oxygen of U9 backbone while Y52 recognizes the N3 of A16 nucleobase (Fig. 3a, b). Both appear to also engage in van der Waals interactions using their aromatic rings. Removing the aromatic moieties by Y50A or Y52A substitution greatly diminished RNA binding, while Y50F and Y52F substitutions produced marginal defects (Fig. 3c, d, Supplementary Fig. 7). Y54 protrudes across the minor groove, packs against the A16 ribose, and appears to engage its O4 group (Fig. 3a, b). Unlike the mutational effects of Y50 and Y52, Y54A conferred little defect while Y54F reduced binding, suggesting a distinct role from Y50 and Y52. Immediately adjacent to Y54, N55 loops back to recognize the 2′-OH group of G10 (Fig. 3a, b). N55A substantially diminished dsRNA binding, indicating the importance of 2′-OH recognition (Fig. 3c, d).

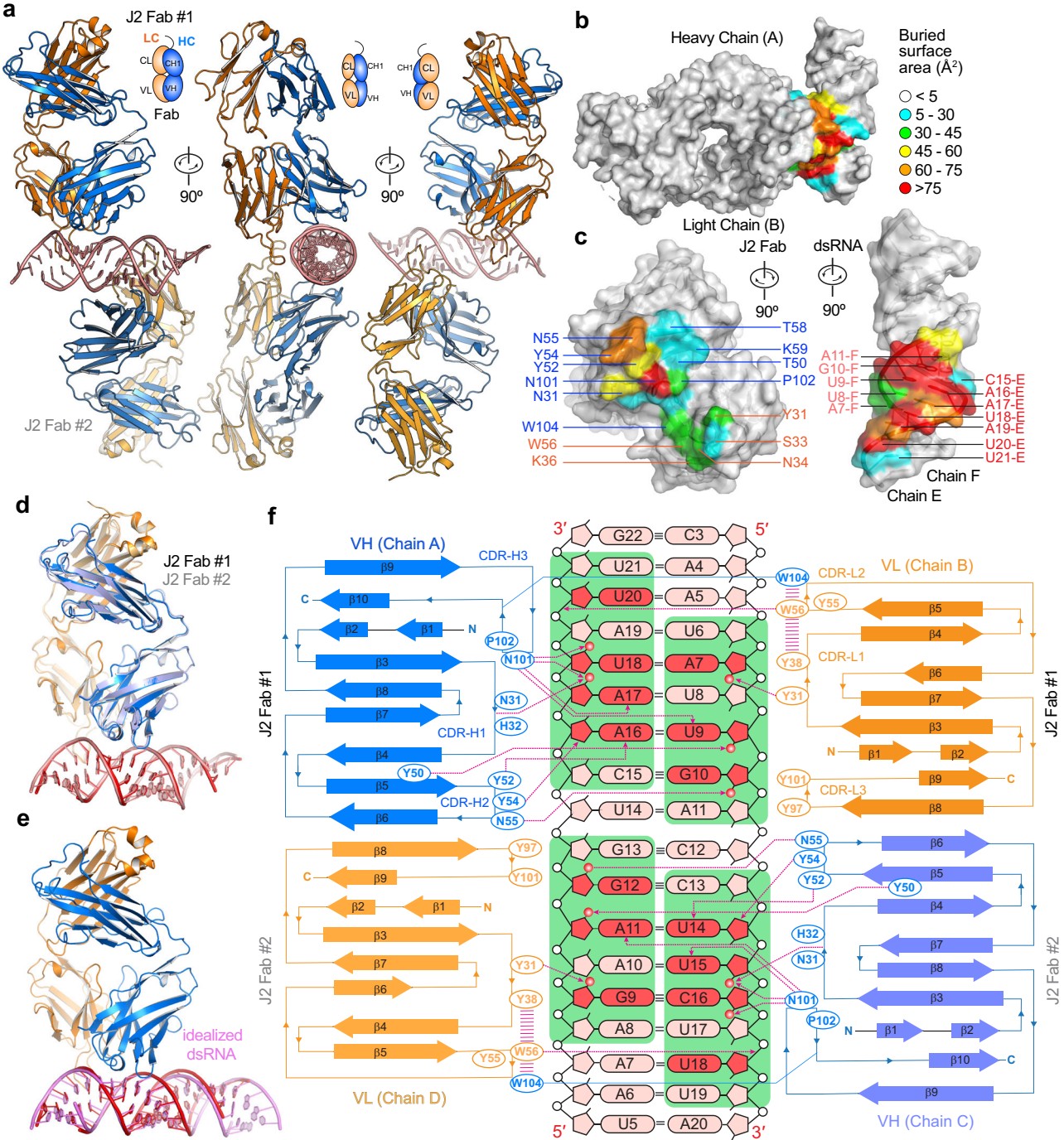

**Fig. 2 | Recognition of dsRNA by J2. a** Front, side, and rear views of the J2 Fab-dsRNA complex structure, showing two Fabs bound to opposite sides of the dsRNA. Heavy chain (HC) is in blue; light chain (LC) in orange. VH and CH1: variable and first constant domains of the heavy chain; VL and CL: variable and constant domains of the light chain. **b, c** Closed (**b**) and open-book (**c**) views of J2 Fab-dsRNA interfaces colored by solvent-accessible areas buried per residue (Å²). **d** Overlay of the two crystallographically observed J2 Fab-dsRNA complexes, colored as in (**a**). **e** Overlay of Fab-dsRNA complex structure colored as in (**a**) with an idealized 23-bp dsRNA (pink). **f** Diagram of secondary structures and interactions of each J2 Fab with the dsRNA. Nucleotides in direct polar contacts are in a darker red color. J2 footprint on the dsRNA is shown in a green background. Proposed hydrogen bonds are shown as dashed magenta lines with arrowheads; π-π stacking interaction as parallel magenta lines. RNA 2′-hydroxyls in contact with J2 are shown as red spheres. Complementarity-determining regions (CDRs) H1-H3 and L1-L3 are denoted.

Buttressed by both CDR-H1 and CDR-H2, CDR-H3 binds near the center of the 8-bp dsRNA epitope, placing its N101 residue equidistant to both RNA strands (Fig. 3a, b). This prime location and flat, groove-parallel orientation of N101 allows it to descend deeper into the minor groove, making five hydrogen bonds to both strands. While the N101 main chain contacts the 2′-OH of U18, its side chain carboxamide concurrently engages the O2 of U9, N3 and 2′-OH of A17. Consistent with these numerous contacts, N101A strongly diminished RNA binding (Fig. 3c). To ask if a basic residue at this location might confer higher levels of RNA binding than the wild type, we replaced N101 with an arginine, which is frequently found to mediate RNA interactions such as forming arginine forks with HIV TAR and 7SK RNAs[26,27].

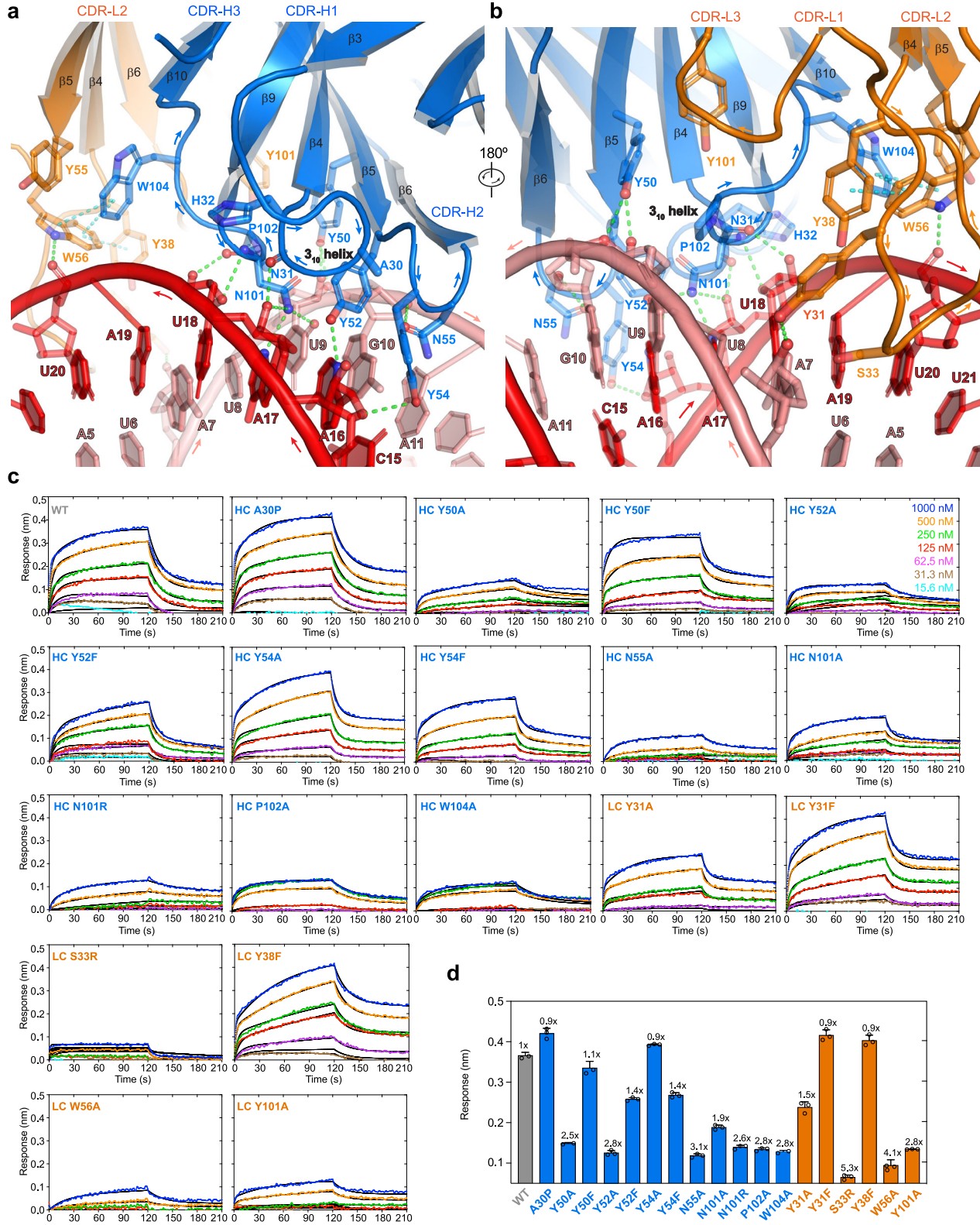

**Fig. 3 | Detailed J2 – dsRNA interfaces. a, b** Detailed interfaces and contacts between the dsRNA (red and salmon, for chains E and F, respectively) and J2 heavy chain CDRs ((**a**), HC, blue) and light chain CDRs ((**b**), LC, orange). Proposed hydrogen bonds and π-π interactions are shown as green and cyan dashes, respectively. For clarity, the 3₁₀ helix in CDR-H1 is shown in cartoon loop instead of helical mode and denoted. **c** Representative BLI sensorgrams for a 20-bp dsRNA binding to WT, HC, and LC mutant J2 IgGs at 100 mM KCl. **d** Summary of the data in (**c**), showing the maximum sensor responses (nm). Values are mean ± s.d. of three biologically independent samples.

Compared to N101A, N101R further compromised RNA binding, suggesting that a bulky basic residue does not make favorable interactions at this location. Immediately adjacent to N101 is P102, which redirects the CDR-H3 loop trajectory and further stacks with the functionally important Y50 (Fig. 3a, b). P102A drastically reduced RNA binding, suggesting its functional importance. W104 closely follow N101 and P102, protrudes into the light chain region, and stacks with W56 of CDR-L2, which in turn stacks with Y38 of CDR-L1 (Figs. 2f, 3a, b). W104 is also well positioned to stack with Y55 of CDR-L2, allowing W104 to stabilize an aromatic network among the LC CDRs. Consistent with this structural role at the HC-LC interface, a W104A substitution severely compromised dsRNA binding (Fig. 3c, d).

Contrasting the extensive contacts made by the heavy chain CDRs, the light chain CDRs L1 and L2 only loosely engage the dsRNA, bind in the periphery of the epitope, and make just a handful of direct contacts. CDR-L3 makes no dsRNA contacts. Specifically, Y31 of CDR-L1 binds the 2′-OH of A7 while W56 of CDR-L2 contacts the phosphate oxygen of U20 (Fig. 3b). Y31A but not Y31F substitution substantially reduced RNA binding, which suggests the importance of hydrophobic interactions with the minor groove over specific hydrogen bonds to the RNA polar groups (Fig. 3a, b, Supplementary Fig. 8). The W56A substitution led to a severe reduction in dsRNA binding, confirming its importance. Considering the relative underutilization of the light chain, we asked if installing a basic residue at the tip of the extended CDR-L2 loop, in the form of S33R adjacent to Y31, might boost RNA binding by J2 (Fig. 3b). However, the S33R substitution nearly abolished RNA binding, presumably due to steric clashes at this snug interface (Fig. 3c, d). The strong negative effects of both HC N101R and LC S33R of J2 contrast with the essential role of HC R104 of the S9.6 antibody in hybrid binding[20]. This suggests that modulating overall electrostatic characteristics of CDRs is not a generally effective strategy to tune antibody binding affinities to duplex grooves. Mirroring the excursion of CDR-H3 W104 to engage with LC CDRs, Y101 of CDR-L3 makes a reciprocal approach to the HC CDRs. It is positioned to make van der Waals interactions with H35 of CDR-H1, W47 of CDR-H2, and F105 of CDR-H3. Due to this putative structural role in stabilizing the HC CDRs, the LC Y101A substitution strongly reduced dsRNA binding, like HC W104A (Fig. 3c, d).

Altogether, each J2 Fab binds 8 bp of dsRNA and recognizes five 2′-OHs, including two consecutive 2′-OHs on each RNA strand across the minor groove (Fig. 2f). Similarly, the S9.6 antibody also recognizes tandem 2′-OHs on the RNA strand[20]. The concurrent recognition of tandem 2′-OHs on both strands by J2 explains its strong preference for dsRNA over RNA-DNA hybrids (Fig. 1c). Numerous aromatic residues form interaction networks both within and between the HC and LC CDRs to stabilize their conformations, which allows the two chains to work in tandem to track a single, extended dsRNA minor groove. Mutational analyses suggest that van der Waals interactions from the aromatic side chains play a larger role in conferring dsRNA binding than individual hydrogen-bonding moieties such as the phenolic hydroxyls. Finally, we evaluated how well AlphaFold 3 can predict the J2-dsRNA interface. Interestingly, AlphaFold 3 successfully predicts the conformation of the J2 Fab, including the CDRs, achieving an RMSD of 0.9–1.0 Å. However, it completely fails to predict the J2-dsRNA interface, placing dsRNA in various orientations and locations far from the CDRs (Supplementary Fig. 9). This finding attests to the remarkable ability of AlphaFold 3 to predict protein structures and conformations, including antibody CDRs, but also highlights its lack of training in unconventional RNA-protein interfaces, such as antibody-nucleic acids interfaces, underrepresented in the PDB.

## J2 exhibits substantial sensitivity to nucleotide composition and skew

Diverse host and viral genomes exhibit wide-ranging GC contents. While the human genome has ~41% GC content, the dsDNA genomes of herpes simplex virus (HSV) contain ~70% GC[28]. By contrast, the SARS-CoV-2 genome has a lower GC content of ~38%. To ask if GC content affects dsRNA recognition by the J2 antibody, we tested a panel of nine 30-bp dsRNAs with varying GC contents, by both FP and BLI (Fig. 4, Supplementary Fig. 10). Both assays reveal that J2 possesses a substantial preference for low- and mid-GC content dsRNAs. Specifically, BLI analyses reveal that while GC contents from 0% up to 63% retained robust J2 binding with $K_d$s in the nanomolar range, dsRNAs that have 77% GC only marginally bound J2, exhibiting micromolar affinities (Fig. 4b, c). Strikingly, dsRNAs that contain 90% or 100% GC were completely unable to bind J2 IgG under the BLI conditions (Fig. 4c, d, Supplementary Fig. 10). This observation is consistent with a previous report that J2 preferred to bind natural, mixed-composition dsRNAs such as the L dsRNA from yeast killer viruses—its original immunogen—over both poly(rA)-poly(rU) and poly(rI)-poly(rC) duplexes[15].

To understand the preference of J2 for nucleotide composition, we first asked whether J2 contacts with specific nucleobase chemical groups could have conferred sequence selectivity. About half of the J2 contacts are to the 2′-OH and phosphate oxygen atoms (Fig. 2f). These RNA backbone contacts are not expected to be sequence selective. The remaining contacts occur between the nucleobases of U9, A16, A17 and side chains of Y52 and N101 of the J2 heavy chain. These are mediated by N3 of purines or O2 of pyrimidines, which are generally indistinguishable between A and G, and between U and C. Therefore, these nucleobase contacts are also unlikely to have produced the observed strong preference in GC content. Next, we explored whether the indirect effects of GC content on the dsRNA helical geometry are responsible for the observed J2 sequence selectivity. GC-rich dsRNAs exhibit substantially stronger intra- and cross-strand stacking, reduced helical rise, and narrower and more rigid minor grooves[29–31]. To ask if the changing GC content had a meaningful impact on the helical geometry and structure of the dsRNAs used in BLI, we employed temperature-scanning Circular Dichroism (CD) analyses of the duplexes and their constituent ssRNA and ssDNA (Figs. 4, 5). RNA CD spectra are complex and multi-component in nature. They include aggregated chiral effects of individual nucleotides such as their glycosidic dihedral angles (*anti* or *syn*), π-π stacking between neighboring nucleobases, and overall helicity of the polynucleotide chains[32–34]. We observed distinct structural features between the AU-rich dsRNAs that bound J2 robustly and their GC-rich counterparts that did not (Fig. 4d–f). Increasing GC content progressively raised the $T_m$ and strengthened intra-strand stacking, as evidenced by the deepening valley at ~210 nm, which was also observed for RNA-DNA hybrids[20]. This is accompanied by the increasing negative ellipticity near ~235 nm. The rising GC content also red-shifted the ~260 nm peak to near ~270 nm, suggesting a change in the A-form helicity due to increased base stacking and exciton coupling[35]. These data suggest that structural changes in the dsRNA helix due to changed nucleotide composition, instead of sequence-specific J2 contacts to nucleobase edges, are likely responsible for their variable compatibilities with J2.

Our FP and BLI data detected robust J2 binding to AU-only duplexes (Fig. 4a–d). However, a previous report noted inefficiency of J2 to detect poly(rA)-poly(rU) duplexes similar to poly(rI)-poly(rC)[15]. To address this discrepancy, we examined J2 binding to five additional 0% GC duplexes (Fig. 5a, Supplementary Fig. 11). We found that J2 exhibited robust binding towards a dsRNA duplex featuring AU dinucleotide steps exclusively with no AU skew between the two strands (Fig. 5a, b). However, binding to the 100% skewed poly(rA)-poly(rU) was much diminished and required high concentrations of the dsRNA. As expected, J2 binding to the poly(rA)-poly(dT) hybrid duplex was even weaker, and there was no binding to the poly(dA)-poly(rU) hybrid or poly(dA)-poly(dT) dsDNA duplexes (Fig. 5b). CD analyses of these duplexes and their constituent ssRNAs exhibited expected spectral features and melting temperatures (Fig. 5b–e). For instance, poly(dA)-poly(dT) dsDNA had characteristic CD peaks at ~280 nm associated

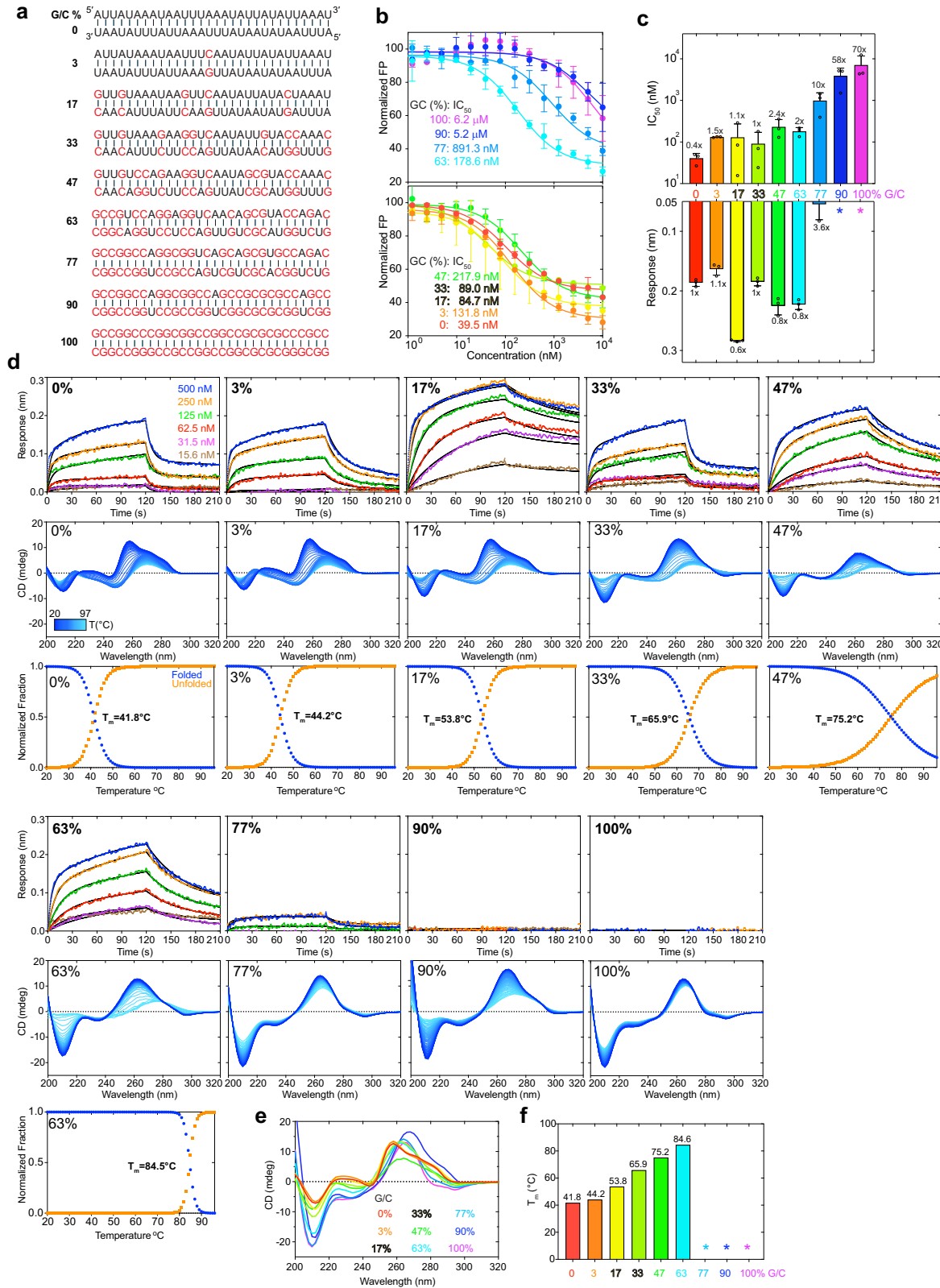

**Fig. 4 | Effects of nucleotide composition on J2 binding. a** Nine 30-bp dsRNAs with GC contents ranging from 0% to 100% used for J2 IgG binding analyses. G-C pairs are indicated in red; A-U pairs in black. **b** Competition FP titrations of dsRNAs in (**a**) against 3′-FAM labeled VA-I RNA prebound to J2 IgG. IC$_{50}$s are indicated. **c** FP-derived IC$_{50}$s derived from (**b**) (top) and BLI maximum sensor responses from (**d**) (bottom) for the dsRNAs in (**a**). Asterisks: no detectable binding. **d** Representative BLI sensorgrams (top, at 200 mM KCl), temperature-scanning CD spectra (middle)

and unfolding transitions (lower) of dsRNAs in (**a**) binding to J2 IgG. Different shades of blue indicate temperature transitions from 20 °C (dark blue) to 97 °C (light blue). **e** Comparison of the CD spectra of dsRNAs in (**a**) at 20 °C. **f** Melting temperatures (T$_m$s) of dsRNAs in (**a**) derived from temperature-scanning CD data in (**d**). Asterisks: T$_m$ > 85 °C and out of measurable range. Values are mean ± s.d. of three biologically independent samples.

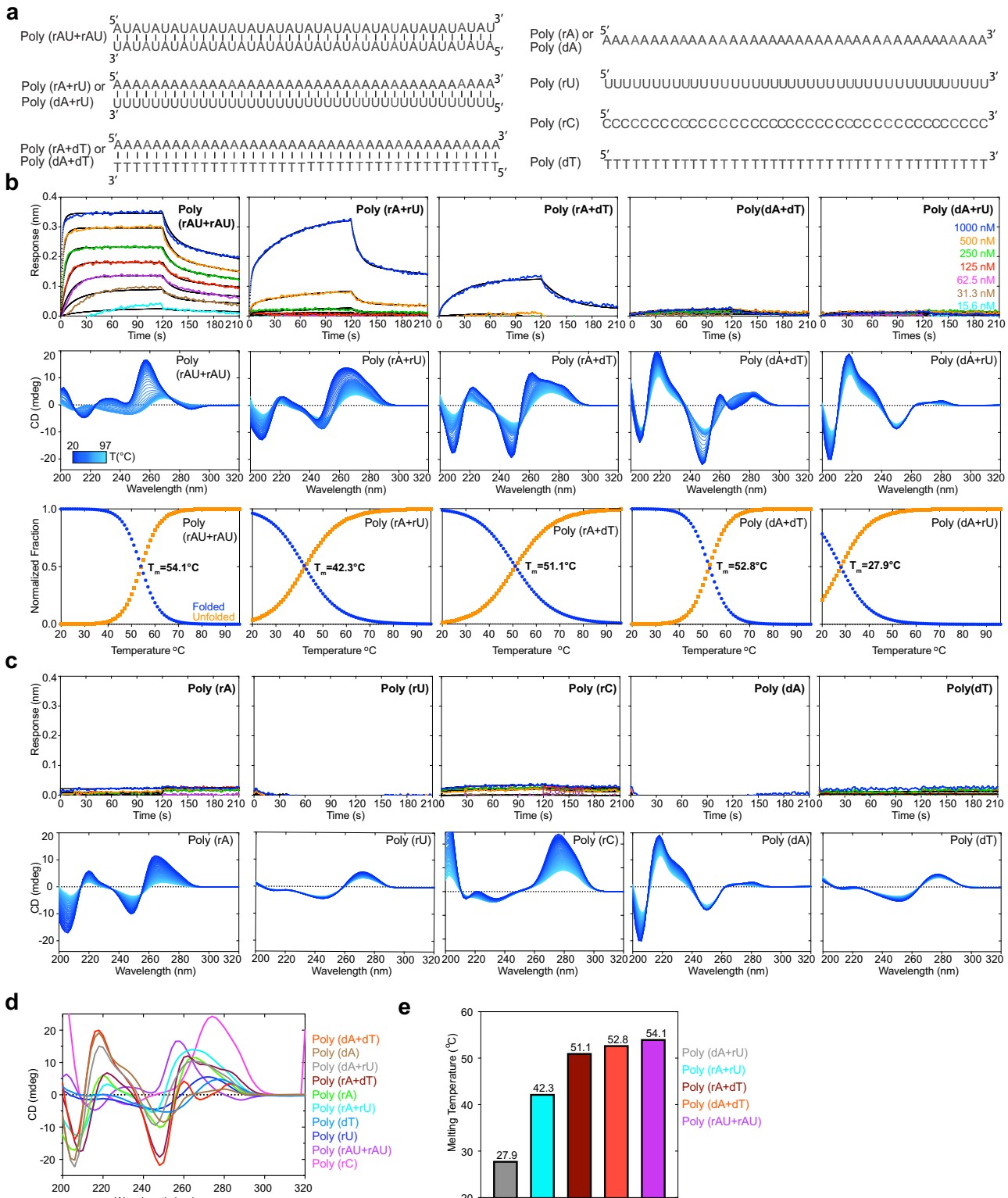

**Fig. 5 | Effects of homotypic repeat sequences and nucleotide skew on J2 binding. a** Sequences of dsRNA, dsDNA, RNA-DNA hybrids, ssRNA, and ssDNA of repetitive sequences (40 bp or 40 nts) used in J2 IgG binding analyses. **b** Representative BLI sensorgrams (top, at 400 mM KCl), temperature-scanning CD spectra (middle) and unfolding transitions (lower) of duplex nucleic acids in (**a**) binding to J2 IgG. Different shades of blue indicate temperature transitions from 20 °C (dark blue) to 97 °C (light blue). **c** Representative BLI sensorgrams (top, at 400 mM KCl) and temperature-scanning CD spectra (bottom) of single-stranded nucleic acids in (**a**) binding to J2 IgG. **d** Comparison of the CD spectra of nucleic acids in (**b**, **c**) at 20 °C. **e** Melting temperatures of duplex nucleic acids in (**a**) derived from temperature-scanning CD data in (**b**).

with B-form geometry, while the poly(rA)-poly(dT) hybrid exhibited mixed A/B character. Compared to the poly(rA)-poly(dT) hybrid, the poly(dA)-poly(rU) hybrid is the least stable duplex due to reduced pairing, stacking, and helical stabilities (Fig. 5e). Consequently, this hybrid facilitates RNA slippage and shearing during transcription termination[36] and reiterative transcription[37].

Together, these data suggest that J2 binding is sensitive to the helical geometry of the dsRNA, which is in turn substantially modulated by their sequence composition and skew. Interestingly, similar conclusions were reached for the S9.6 antibody, which preferably binds higher GC content hybrids[20].

## J2 effectively recognizes the heterogeneous structure of poly(rI)-poly(rC)

Next, we evaluated J2 recognition of poly(rI)-poly(rC), which is widely used as a synthetic dsRNA analog that robustly activates Toll-like receptor 3 (TLR3), PKR, and other dsRNA sensors[38,39]. Notably, commercial high-molecular-weight (HMW) poly(rI)-poly(rC) is of variable lengths ranging from 1.5–8 kb and consists of annealed poly(rI) and poly(rC) strands. Due to its homotypic sequences and imperfect annealing of such long strands, poly(rI)-poly(rC) is known to exhibit a highly heterogeneous structure containing a mixture of dsRNA and ssRNA segments[40,41]. We performed temperature-scanning CD analysis of poly(rI)-poly(rC), which reveals unique spectra distinct from both AU-rich and GC-rich dsRNAs but nearly identical to previous measurements (Supplementary Fig. 12). Specifically, poly(rI)-poly(rC) spectra are characterized by prominent 245 and 280 nm peaks, suggesting unusual exciton coupling from the stacked inosine-cytosine pairs and an altered helical structure. This unusual helical geometry is likely at least partially responsible for the previously reported diminished affinity with J2[15,18]. We observed a relatively low $T_m$ of ~55.7 °C, also comparable with previous reports. Considering its average length of ~4.8 kb, this low $T_m$ is consistent with poly(rI)-poly(rC) duplexes being regularly interrupted by ssRNA bulges and loops that reduce the $T_m$. We then monitored poly(rI)-poly(rC) binding to J2 IgG using BLI. Robust signal was observed at even picomolar concentrations of poly(rI)-poly(rC) (Supplementary Fig. 12). However, the bound poly(rI)-poly(rC) exhibited no discernible dissociation from J2 IgG, presumably due to cross-bridging of multiple J2 IgGs contacting and retaining the same lengthy poly(rI)-poly(rC) molecule. We conclude that J2 is competent in detecting poly(rI)-poly(rC), but it is difficult to quantitatively compare its binding affinity with perfectly paired dsRNA due to its heterogeneous and presumably branched structure. Notably, in typical immunofluorescence experiments with cells, J2 IgG is not immobilized and thus not expected to exhibit extensive cross-bridging with poly(rI)-poly(rC) beyond possible IgG bivalency driven by both Fabs engaging at the same time.

## Effects of dsRNA length and Mg$^{2+}$ on RNA helical geometry

The crystallographically observed 8-bp footprint of J2 Fab on dsRNA and its preferred binding to dsRNA 30 bp or longer may seem at odds with each other. To understand why shorter dsRNAs exhibit diminished binding to J2 despite in theory containing a full epitope, we examined their helical geometry using temperature-scanning CD. Interestingly, we found that as dsRNAs increased in length, their CD spectra progressively changed, concomitant with an expected increase in $T_m$ (Supplementary Fig. 13). The ~276 nm peak typically associated with B-form helix gradually blue-shifted to ~260 nm while the ~210 nm valley deepened, suggesting length-dependent strengthening of A-form helical conformation and intra-strand stacking, respectively. These data indicate that short dsRNA duplexes don't exhibit the full A-form helical structure of long dsRNAs, presumably due to insufficient accumulative coaxial stacking that compresses the dsRNA and reduces the helical rise. In addition, base pairs near helical termini are prone to fraying and transient opening, further reducing the length of

stable dsRNA segment[26,27]. Therefore, the altered minor groove geometry of short dsRNAs is not well recognized by J2, whose original immunogen was a ~4.6 kb long viral dsRNA. An alternative explanation is that longer dsRNA is more likely to engage bivalent IgG binding via its two Fab arms. However, there is substantial steric hindrance that prevents the two Fab arms of IgG from recognizing immediately juxtaposed or contiguous epitopes. Recent analyses using DNA origami and other synthetic antigen arrays revealed that bivalent IgG binding prefers a 100–150 Å spacing (36–58 bp dsRNA) between adjacent epitopes[42–45]. Further, the rigidity of dsRNA imposes additional orientation requirements on both J2 Fabs in order to achieve IgG bivalency. These considerations and our AUC and SEC-MALS data favor the notion that shorter dsRNA segments exhibit diminished J2 binding due to their altered helical geometry as opposed to a lack of bivalent IgG binding.

Finally, we examined how Mg$^{2+}$ may modulate the dsRNA helical structure[46–48] and indirectly affect J2 interactions. We collected temperature-scanning CD spectra of 30-bp dsRNAs of low (0%), mid (47%), and high (100%) GC contents, at 0, 1, and 2 mM Mg$^{2+}$. As expected, Mg$^{2+}$ stabilized the A-form helical geometry of low- and mid-GC dsRNAs and raised their $T_m$s, while exerting minimal effects on the high-GC dsRNA that is already stable without Mg$^{2+}$ (Supplementary Fig. 14). There was little difference in the CD spectra collected at 1 or 2 mM Mg$^{2+}$. Together, these data suggest that RNA topology (e.g., poly(rI)-poly(rC)), dsRNA length, and solute conditions (including monovalent and divalent cations such as Mg$^{2+}$) likely additionally impact J2 interactions.

## Duplex-binding antibodies employ distinct global and local strategies from intracellular dsRNA-binding proteins

The shared strategy of helical, minor groove-tracking used by J2 and S9.6 as secreted antibodies contrasts starkly with known strategies employed by most intracellular dsRNA-binding proteins (Fig. 6a). For instance, dsRBMs bind across three consecutive grooves (minor, major, minor) on the same face, and linearly track the helical axis of dsRNA (Fig. 6b)[49]. Similarly, transmembrane dsRNA channel SID1 linearly engages four adjacent grooves to capture and import dsRNAs[50]. By contrast, both J2 and S9.6 limit their contacts to a single minor groove and use their heavy and light chain CDR clusters in tandem to make continuous contacts along the extended groove. The free S9.6 antibody structure was nearly identical to that of the hybrid-bound structure, suggesting conformational rigidification of the CDRs prior to binding[20]. Such relatively rigid paratopes, common for affinity-matured antibodies, are expected to be less adaptable towards significant variations in the minor groove geometry and chemical environment due to nucleotide composition, skew, and duplex bending. Therefore, the groove-tracking modality is likely intrinsically less tolerant towards divergent duplex sequences, geometries and local deformations. This notion explains the divergent sequence preferences observed for different dsRNA antibodies, where J2 and J5 preferred mixed composition while K1 exclusively bound poly(rI)-poly(rC)[15]. By contrast, axis-tracking, ridge-spanning intracellular domains such as dsRBMs may be more tolerant of groove width variations by focusing on the recognition of the phosphodiester backbone and 2′-OHs and deploying flexibly tethered repeat domains that jointly conform to long, deformable dsRNAs[51].

Besides the differences in overall topology, antibodies and intracellular proteins also favor distinct types of chemical contacts mediated by dissimilar amino acid types. Both J2 and S9.6 employ extensive networks of aromatic residues (e.g., Tyr and Trp) to bind the minor groove hydrophobic features (portions of nucleobases and riboses) through van der Waals interactions. This is evidenced by the functional sufficiency of Phe side chains in place of Tyr at LC Y31, HC Y50 and Y52 positions (Fig. 3c), and the essential HC Y101 position of the S9.6 antibody[20]. For comparison, multiple synthetic antibodies against

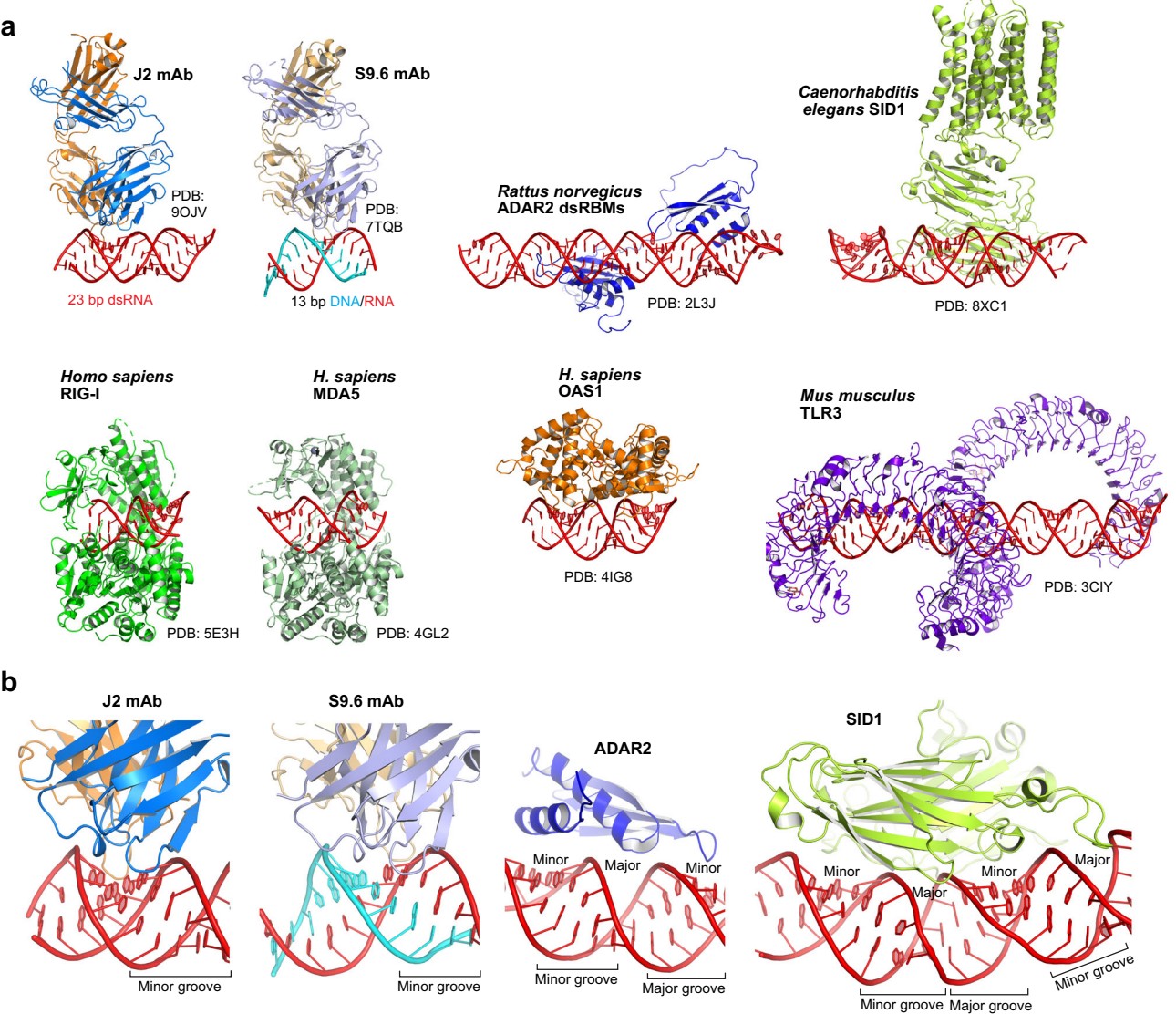

**Fig. 6 | Comparison of dsRNA recognition by secreted antibodies *versus* intracellular proteins. a** Overall structures of J2 and S9.6 antibodies bound to dsRNA and RNA-DNA hybrid, respectively, and dsRNA-bound structures with ADAR2, SID1, RIG-I, MDA5, OAS1, and TLR3. **b** Comparison of the minor groove tracking strategy employed by J2 and S9.6 with the helical axis tracking, ridge-spanning strategy used by ADAR2 and SID1.

structured RNAs discovered through phage display also extensively leverage non-polar interactions in addition to polar contacts[52–58]. By contrast, dsRBMs primarily utilize strategically positioned polar residues (Lys, Arg, His, Gln, Asn, Glu) to bind backbone chemical groups, including 2′-OHs and phosphate oxygens[49,59,60]. These two distinct approaches to dsRNA recognition likely evolved independently to better suit the local chemical environments of the interfaces and the characteristics of the protein folds being employed.

## Discussion

The main findings of this work are: (1) the widely used J2 monoclonal antibody is highly specific for dsRNA. (2) J2 can detect dsRNAs as short as ~14 bp but binding strength increases with dsRNA length. (3) J2 recognition is sensitive to nucleotide composition and skew due to their effects on helical geometry. (4) J2 uses its heavy and light chain CDR clusters in tandem to track the dsRNA minor groove, employing both aromatic and polar residues to recognize both strands of a 2-nt-staggered 8-bp duplex.

For three decades, the J2 antibody has been an essential tool to discover and characterize novel RNAs in new locations, and to probe viral particle composition and replication mechanisms. For instance, extracellular RNAs (exRNAs) are being rapidly discovered, ranging from glycosylated tRNAs, vault RNAs, and lncRNAs on cell surfaces to viroid-like "obelisks" in human microbiomes[61–63]. Their discovery, experimental detection and validation have largely relied on the J2 antibody. Little information was heretofore available regarding the specificity, epitope, and mechanisms of this antibody, or other, less-used dsRNA antibodies, including K1, K2, J5, and 9D5[15,64]. This gap in knowledge precludes reliable interpretation of numerous immuno-fluorescence, immunoblotting, immunohistochemistry, immunopre-cipitation sequencing (IP-Seq), ELISA, and flow cytometry analyses in ongoing studies using this antibody. Our data demonstrate that the J2 antibody is a highly selective tool that does not substantially cross-react with other types of nucleic acids, including dsDNA, RNA-DNA hybrids, ssRNA and ssDNA. Its remarkable ability to reject RNA-DNA hybrids ensures that J2 does not substantially bind cellular R-loops or other exposed hybrids. By contrast, the R-loop-specific S9.6 antibody exhibits substantial cross-reactivity with dsRNA, rendering it ineffectual for intracellular R-loop imaging[20,23]. The practical detection threshold of ~14-bp allows J2 to avoid interactions with abundant

cellular tRNAs and other small RNAs, but effectively detect naturally occurring, medium-sized viral dsRNAs such as Adenovirus VA-I.

Our finding that J2 recognizes dsRNA as short as 14 bp contrasts with the previous estimate of ~40 bp[18]. This substantially expands the utility of J2 in potentially detecting numerous smaller dsRNAs, such as miRNAs and siRNA precursors, which are more than 20 bp in length. It may also enable the use of J2 in mapping dsRNA segments present on larger RNAs such as lncRNAs, mRNAs, and viral RNA genomes. This potential new application alleviates the current unavailability of the RNase V1, conventionally used to positively identify dsRNA segments.

A limitation of the J2 antibody is its diminished ability to detect dsRNAs of high-GC contents, such as those derived from the genomes of Herpesviruses, Poxviruses, Hepatitis C/E/G, etc[65]. Previous analyses relying on J2 detection or enrichment could benefit from data reassessments as to whether high-GC dsRNAs may have been disproportionately filtered or inadvertently excluded. To detect or enrich high-GC dsRNAs, alternative dsRNA antibodies may prove more effective than J2, although their sequence specificities have not been systematically assessed. Contrasting J2, the S9.6 antibody preferably binds high-GC duplexes[20]. Given its substantial cross-reactivity with dsRNA, S9.6 could potentially be used to detect high-GC dsRNAs instead of J2. Bifunctional antibodies that combine a lower-GC-preferring J2 Fab and a higher-GC-preferring S9.6 Fab may help mitigate their opposing sequence biases and recognize dsRNAs across a wide range of nucleotide compositions.

Our finding that secreted antibodies and intracellular proteins adopt distinct strategies in dsRNA recognition suggests that at least two independent pathways have naturally evolved for achieving affinity and specificity at dsRNA-protein interfaces. Both modalities have context-dependent advantages and disadvantages, and are not mutually exclusive. While intracellular RNA-binding proteins function in the nucleoplasm and cytoplasm, antibodies first undergo affinity maturation in germinal centers inside lymph node follicles, and then function in blood plasma, lymph, and interstitial fluids. Both the germinal centers and extracellular fluids have higher ionic strengths than the cytoplasm (~150 mM versus ~100 mM), typically contain $Na^+$ ~140 mM, $K^+$ ~4 mM, $Ca^{2+}$ ~2 mM, $Mg^{2+}$ ~1 mM, $Cl^-$ ~110 mM, $HCO_3^-$ ~25 mM, and are mildly oxidizing[66]. Importantly, they primarily contain small, fully dissociated ions ($Na^+$, $Cl^-$, $HCO_3^-$) that contribute strongly to ionic strength, which effectively suppresses nonspecific electrostatic interactions. This makes antibodies that leverage hydrophobic interactions more effective and specific for nucleic acid interactions. By contrast, intracellular fluids are $K^+$-rich and $Na^+$-poor, and contain mostly large organic anions (ATP, proteins, nucleic acids) that are not fully dissociated and less mobile. Such conditions favor dynamic electrostatic interactions between proteins (e.g., basic residues) and nucleic acids due to the reduced shielding of their charges. Despite their advantages, antibodies are limited by the pre-determined physical dimensions of CDR regions and the conformational and steric constraints between the two Fabs[67]. Antibodies also cannot deploy flexibly tethered, fully articulating domain arrays exemplified by tandem dsRBMs to recognize larger, conformationally flexible epitopes such as stretches of dsRNA[51]. Engineered antibody-derived protein assemblies, such as tethered single-chain variable fragments (scFv) and nanobodies, could combine the benefits of both modalities and achieve superior dsRNA binding affinity and selectivity.

## Methods

### Sequences and expression plasmids for J2 IgG and J2-Fab
The protein sequences for J2-IgG HC and LC were derived from a previous report[68]. The sequences of the J2 antibody from the hybridoma cells were confirmed using LC-MS/MS. The DNA sequences for J2-IgG HC, J2-Fab HC and J2-IgG LC were codon-optimized for expression in CHO cells and synthesized as gBlocks (Integrated DNA Technologies, Coralville, IA). The HC-IgG sequence was extended at the 3'

end to encode a GGG linker followed by an octa-histidine tag (His8). The HC-Fab sequence was extended at the 3' end to encode a sortase signal ("SoSi", LPETGG) and followed by a hexa-histidine tag (His6). The ends of the gBlocks included XbaI (5') and HindIII (3') restriction sites with which the individual gBlocks were cloned into XbaI/HindIII cleaved expression vector pcDNA3.4 to generate pcDNA3.4-J2-IgG_HC-TEV-His8, pcDNA3.4-J2-Fab_HC-SoSi-His6 and pcDNA3.4-J2-IgG_LC. Amino acid substitutions were generated using the Q5 Site-Directed Mutagenesis Kit (NEB). Plasmid sequences and mutations were confirmed by Sanger sequencing and whole plasmid sequencing (Psomagen, Rockville, MD). Amino acid sequences of the resulting J2-IgG HC, J2-Fab HC and J2-IgG LC are given below, where signal sequences (removed during secretion) are underlined.

**J2-IgG HC-TEV-His8.** <u>MGWSCIILFLVATATGVHS</u>QVQLQQSGPELVKP GASVKMSCKASGYTFANHVMHWVKQKPGQGLEWIGYIYPYNDGTKYN EKFKGKATLTSDKSSSTAYMELSSLASEDSAVYYCARGGNPAWFAYWGQ GTLVTVSAAKTTAPSVYPLAPVCGDTTGSSVTLGCLVKGYFPEPVTLTWN SGSLSSGVHTFPAVLQSDLYTLSSSVTVTSSTWPSQSITCNVAHPASSTKV DKKIEPRGPTIKPCPPCKCPAPNLLGGPSVFIFPPKIKDVLMISLSPIVTCVV VDVSEDDPDVQISWFVNNVEVHTAQTQTHREDYNSTLRVVSALPIQHQD WMSGKEFKCKVNNKALPAPIERTISKPKGPVRAPQVYVLPPPEEEMTKKQ VTLTCMVTDFMPEDIYVEWTNNGKTELNYKNTEPVLDSDGSYFMYSKL RVEKKNWVERNSYSCSVVHEGLHNHHTTKSFSRTPGKGGGHHHHHHHH.

**J2-Fab HC-SoSi-His6.** <u>MGWSCIILFLVATATGVHS</u>EVQLQQSGPELVKP GASVKMSCKASGYTFANHVMHWVKQKPGQGLEWIGYIYPYNDGTKYNE KFKGKATLTSDKSSSTAYMELSSLASEDSAVYYCARGGNPAWFAYWGQG TLVTVSAAKTTAPSVYPLAPVCGDTTGSSVTLGCLVKGYFPEPVTLTWNS GSLSSGVHTFPAVLQSDLYTLSSSVTVTSSTWPSQSITCNVAHPASSTKVD KKISALPETGGGHHHHHH.

**J2-IgG LC.** <u>MGWSCIILFLVATATGVHS</u>NIMMTQSPSSLAVSAGEKVTM SCKSSQSVLYSSNQKNYLAWYQQKPGQSPKLLIYWASTRESGVPDRFTGS GSGTDFTLTISSVQAEDLAVYYCHQYLSSYTFGGGTKLEIKRADAAPTVSIF PPSSEQLTSGGASVVCFLNNFYPKDINVKWKIDGSERQNGVLNSWTDQD SKDSTYSMSSTLTLTKDEYERHNSYTCEATHKTSTSPIVKSFNRRNEC.

### Expression and purification of J2-IgG and J2-Fab in ExpiCHO-S cells
The ExpiCHO Expression System kit (ThermoFisher Scientific, Waltham, MA) was used according to manufacturer-recommended protocols with modifications. Briefly, ExpiCHO-S cells were grown in ExpiCHO expression medium in plain-bottom Erlenmeyer flasks with vented screw caps at 37 °C and 8% $CO_2$ with continuous shaking at 120 rpm. ExpiCHO cells having viability > 95% were transfected at a cell density of $5 \times 10^6$ cells/mL. Each 100-ml portion of expiCHO cells was transfected with 50 μg of each of IgG HC and IgG LC or Fab HC and IgG LC plasmids mixed with 8 mL of OptiPRO SFM and 320 μL of ExpiFectamine CHO reagent. These were combined and incubated for 1–5 min before addition to the cells. Following the "Max Titer" protocol, 600 μl ExpiCHO Enhancer and 16 ml of ExpiCHO Feed were added 24 h post-transfection. The cells were moved to an incubator shaker kept at 32 °C and 5% $CO_2$, 120 rpm. An additional 16 ml ExpiCHO Feed was added on day 5, and the culture was harvested on days 10-12.

J2-IgG and J2-Fab proteins were purified from culture supernatant using their C-terminal histidine tags. Following centrifugation to remove the expiCHO cells, the supernatant was incubated batchwise with gentle rolling at 5 °C for at least 1 h with 5% (w/v) Ni-NTA agarose beads (Qiagen). This amount of Ni-NTA agarose exceeds that normally used because materials in the medium interfere, possibly by chelation of the nickel. The agarose beads were collected on a porous funnel or column, washed with 30 mM Imidazole, 200 mM NaCl, pH 7.0, and the Fab proteins were then eluted with 300 mM Imidazole, pH 7.0. The purified proteins were dialyzed in 10 mM Tris (pH 7.2) and 100 mM

NaCl and further purified by size-exclusion chromatography on a Superdex 200 column (Cytiva). Protein purity was assessed by SDS-PAGE on 4–20% Tris-Glycine polyacrylamide gels (Invitrogen). For reduced samples, the J2-IgG and J2-Fab proteins were treated with 1 mM dithiothreitol (DTT). To verify the presence of intended mutations, all Fab proteins were reduced by DTT and analyzed by mass spectrometry.

## RNA preparation

tRNA used in biophysical analyses was prepared by T7 RNA polymerase in vitro transcription using PCR-generated templates[69–71]. To reduce the N + 1 extension activity of T7 RNAP, two consecutive 2′-O-methyl modifications were introduced on the 5′ ends of the template strands. RNAs were purified by denaturing gel electrophoresis in Tris-borate EDTA buffer on 10% polyacrylamide (29:1 acrylamide:bisacrylamide) gels containing 8 M urea. RNAs were eluted by overnight crush and soak in 300 mM NaOAc, pH 5.2, and 1 mM EDTA at 4 °C, washed with 1 M KCl using Amicon centrifugation filters, followed by two washes with DEPC-treated water. The concentrated RNAs were filtered and stored at −20 °C until use. Oligonucleotides used for BLI and fluorescence polarization assays were purchased from Integrated DNA Technologies or Horizon Discovery Dharmacon and used without further purification.

## Bimolecular fluorescence polarization (FP) analysis

For bimolecular FP assays, 5 nM 3′ FAM-labeled Adenovirus VA-I RNA (Supplementary Fig. 2) was titrated with increasing amounts of J2 IgG in a buffer consisting of 25 mM Tris-HCl (pH 7.5), 100 mM KCl, and 2 mM MgCl$_2$ in a 96-well plate at 21 °C. FP values were measured in triplicate using a BMG CLARIOstar Plus microplate reader with excitation at 482 nm, emission at 530–540 nm, and LP (long pass) 504 dichroic filter setting. Changes in FP as a function of J2 IgG concentrations were fit with the following equation to determine the apparent dissociation constant $K_d$ using GraphPad Prism 10.

$$Y = B_{max} \times X / (K_d + X). \tag{1}$$

## Competition fluorescence polarization (FP) analysis

For competition FP assays, 10 nM 3′ FAM-labeled Adenovirus VA-I RNA was first mixed with 160 nM J2 IgG and kept on ice for 10 min to form a complex. Competing dsRNAs were heated at 90 °C for 3 min in 25 mM Tris-HCl (pH 7.5) and 100 mM KCl, and rapidly cooled over 2 min, followed by the addition of 2 mM MgCl$_2$. The nucleic acids were then serially diluted, and 20 μL of each diluted solution was combined with 20 μL of complex solution in a buffer composed of 25 mM Tris-HCl (pH 7.5), 100 mM KCl, and 2 mM MgCl$_2$ in a 96-well plate, and incubated at 21 °C for 10 min. FP values were recorded using the same settings as for the bimolecular binding measurements above. Changes in FP as a function of competitor concentrations were fit to the following model to determine the half-maximal inhibitory concentration (IC$_{50}$):

$$Y = Bottom + (Top - Bottom)/(1 + 10^{(X - LogIC_{50})}) \tag{2}$$

## Biolayer Interferometry (BLI)

Gator Bio Protein A (ProA) probes were first hydrated for 10 min in a BLI assay buffer composed of 25 mM Tris-HCl (pH 7.5), 200 mM KCl, and 2 mM MgCl$_2$. Assays were performed in triplicate using flat black-bottom 96-well plates, with each well consisting of 200 μL of buffer, ligand, or analyte solution arranged by column. Columns 1 and 3 contained BLI assay buffer for baseline measurements; column 2 contained the ligand solution (30 μg/mL of J2 IgG) for probe loading; column 4 contained the analyte solution (nucleic acids). The analytes

were serially diluted by a factor of two into the wells of column 4 with a starting concentration of 1 μM. The sensors were systematically transferred through a 5-step protocol: column 1 for baseline, column 2 for ligand loading, column 3 for second baseline, column 4 for analyte association and dissociation back in column 3. Each step was performed at 25 °C for 120 s with an orbital shaking speed of 1000 rpm. Data analyses and global fits were conducted using the Gator Bio software provided by the manufacturer. For each sensorgram set, fits to the homogeneous and heterogeneous ligand models were performed and compared. Essentially, all dsRNA data fit the heterogeneous ligand model only, while the VA-I RNA data fit the homogeneous ligand model. Reduced orbital speeds of 200, 400, and 800 rpm were tested with a 30-bp dsRNA, which produced similar sensorgrams as 1000 rpm and could only be fit with the heterogeneous ligand model. Due to the steep dependency of dsRNA binding strength to J2 IgG on ionic strength, KCl concentrations were adjusted (noted in figure legends where applicable) so that the sensorgrams capture sufficient association and dissociation signal. Kinetic parameters ($k_{on}$, $k_{off}$), kinetics-derived $K_d$ ($k_{off}/k_{on}$), and steady-state $K_d$ values are in Supplementary Table 1. $K_d$ (steady state) was derived by curve-fitting sensor responses (nm) as functions of dsRNA concentrations. Global fits using the heterogeneous ligand model suggest an approximately equal split between the two observed binding components for dsRNA. Only the parameters of the stable, concentration-dependent component were reported, as is implemented in the manufacturer's provided software. The parameters for the other component were generally unstable, out of range, and could not be accurately determined. They may be associated with nonspecific surface phenomena not associated with the productive J2-dsRNA interaction under study. $K_d$s derived from steady-state analyses generally match those derived from kinetic analyses, which indicate reliable measurements in both modalities (Supplementary Table 1).

## Temperature-scanning circular dichroism (CD)

CD experiments were performed using 10 μM of nucleic acids in 25 mM Tris-HCl, pH 7.5, 25 mM NaCl, and 2 mM EDTA on an Applied Photophysics Chirascan Q100 spectropolarimeter. The CD spectra were recorded from 195 to 320 nm using a 1 mm pathlength cell. The temperature range was from 20 to 97 °C with a step size of 1 °C min$^{-1}$. The CD data were analyzed using the Global3 software provided with the instrument by Applied Photophysics. Melting temperatures (T$_m$s) were determined by global fits of all the spectra across the temperature range.

## Crystallization and structure determination of the J2 Fab-dsRNA complex

The complex was reconstituted at 6 mg/mL by mixing the J2 Fab with a 24-bp dsRNA in 25 mM Tris-HCl, pH 7.5, 50 mM NaCl and 1 mM MgCl$_2$. The dsRNA contains a terminal AxA mismatch. The sequences were 5´-GUUGUAAAGAAGGUCAAUAUUGUA-3´ (Chain E) and 5´-AACAAUAUUGACCUUCUUUACAAC-3´ (Chain F). Crystals were obtained by sitting drop vapor diffusion and grew over two weeks at 20 °C, by mixing 1:1 the complex with a reservoir solution composed of 0.15 M malic acid, pH 7, 0.1 M imidazole and 22% w/v PEG MME 550. Crystals were cryoprotected in the same reservoir solution supplemented with 30% ethylene glycol and vitrified at 100 K. Diffraction data were collected at SER-CAT beamline 22-ID at the Advanced Photon Source (APS). X-ray diffraction data were indexed, integrated, and scaled using XDS[72]. Phase information was obtained by molecular replacement using Phaser[73] and the S9.6 Fab structure (PDB: 7TQB) and an 18-bp dsRNA generated in Coot[74] as search models. Iterative rounds of model building were performed in Coot[74] and refined using Phenix.Refine[75]. X-ray crystallographic data collection and refinement statistics are summarized in Table 1.

## Size-exclusion chromatography coupled to multi-angle light scattering (SEC-MALS)

To measure the stoichiometry of J2 antibody binding to dsRNA, 40 µM of J2 Fab was mixed with 40 µM of 27-bp dsRNA for 10 min at room temperature in a buffer consisting of 25 mM Tris-HCl, pH 7.5, 50 mM NaCl and 1 mM $MgCl_2$, prior to injection onto a Superdex 200 Increase column (Cytiva). The column was equilibrated in 25 mM Tris-HCl, pH 7.5, 50 mM NaCl and 1 mM $MgCl_2$ and connected to an Agilent HPLC system. The HPLC system was coupled to a DAWN HELEOS II detector equipped with a quasi-elastic light scattering module and an Optilab T-rEX refractometer (Wyatt Technology). Data were analyzed using the ASTRA 7.3 software (Wyatt Technology Europe).

**Sedimentation velocity analytical ultracentrifugation (AUC).** Sedimentation velocity experiments were carried out at 50,000 rpm (201,600 x g at 7.20 cm) and 20 °C on a Beckman Coulter ProteomeLab XL-I analytical ultracentrifuge and An50-Ti rotor following standard protocols[76]. Samples of 5 µM J2 Fab, 5 µM 23 bp dsRNA and their mixtures were prepared for analytical ultracentrifugation in 50 mM NaCl, 25 mM Tris-HCl, pH 7.5, and 1 mM $MgCl_2$. Samples of 1 µM and 2 µM J2 IgG, 1 µM 30 bp dsRNA (17% GC), and their mixtures were prepared for analytical ultracentrifugation in 100 mM KCl, 50 mM Tris-HCl, pH 8.0, and 2 mM $MgCl_2$. Sedimentation data were analyzed in SEDFIT[77] in terms of a continuous c(s) distribution of sedimenting species. The solution density, viscosity, and protein partial specific volumes were calculated based on their composition in SEDNTERP[78]. A partial specific volume of 0.50 $cm^3 g^{-1}$ was used for RNA. Additivity rules were used to determine the partial specific volumes for the complexes. Because of differences in the partial specific volumes, experimental sedimentation coefficients present the c(s) distributions.

## Reporting summary

Further information on research design is available in the Nature Portfolio Reporting Summary linked to this article.

# Data availability

Atomic coordinates and structure factor amplitudes for the J2 Fab in complex with a dsRNA have been deposited at the Protein Data Bank under accession code 9OJV. Other atomic coordinates used include S9.6 Fab (PDB: 7TQB). Source data are provided with this paper.

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

## Acknowledgments

We thank I. Botos for computational support, D. Wu and G. Piszczek and for support in biophysical analyses, and I. Skeparnias, A.-N. Shaukat, for discussions. This research was supported by the Intramural Research Programs of the National Institute of Diabetes and Digestive and Kidney Diseases (NIDDK) (ZIADK075136 to J.Z.) and National Institute of Allergy and Infectious Diseases (NIAID) (ZIAAI000929 to S.H.L.), an NIH Deputy Director for Intramural Research (DDIR) Challenge Award to J.Z. within the National Institutes of Health (NIH), and the Center for Structural Biology of HIV-1 RNA (CRNA) supported by NIAID U54 AI17660. The contributions of the NIH author(s) are considered Works of the United States Government. The findings and conclusions presented in this paper are those of the author(s) and do not necessarily reflect the views of the NIH or the U.S. Department of Health and Human Services. C.B.N. is a recipient of an Intramural AIDS Research Fellowship (IARF), a NIDDK Nancy Nossal Fellowship Award, and an NIAID Pathway to Independence (K99/R00) Award. This research used resources of the Advanced Photon Source, a U.S. Department of Energy (DOE) Office of Science user facility operated for the DOE Office of Science by Argonne National Laboratory under Contract No. DE-AC02-06CH11357.

## Author contributions

C.B.N., S.H.L., and J.Z. conceived and designed the work. A.B., S.H.L., and D.N.G. generated the mutant libraries, expressed and purified all proteins. C.B.N. performed SEC-MALS analysis, prepared the crystals, collected diffraction data and determined the structure. C.B.N. and J.Z. analyzed the structure. K.M.J., A.J.B., and C.B.N. performed BLI and FP analyses. K.M.J. performed CD analyses. R.G. performed AUC analyses. M.S. performed protein sequencing. All authors contributed to data interpretation and manuscript preparation.

## Funding

## Competing interests

The authors declare no competing interests.
