## [Transparent Peer Review file · Nature Communications]

Structural basis of double-stranded RNA recognition by the J2 monoclonal antibody

Corresponding Author: Dr Jinwei Zhang

Version 0:

Reviewer comments:

Reviewer #1

(Remarks to the Author)

The manuscript by C. Bou Nader et al. provides a comprehensive and thorough analysis of the J2 monoclonal antibody which is used to detect endogenous and exogenous dsRNA. The study established J2 as a double stranded RNA (dsRNA) specific antibody that discriminates against other single stranded nucleotides and nucleotides of mixed composition. The mechanism of epitope binding by J2 antibody was revealed through binding experiments with various lengths of dsRNA and the co-crystal structure of the J2 antibody fragment with a dsRNA. The study expands the utility of J2 antibody as a dsRNA detection tool by determining the minimum required length of dsRNA for antibody binding while also highlighting the limitations of the tool by showing its strong preference for low GC content dsRNA. The study increases the reliability of dsRNA detection by J2 antibody.

The co-reviewers do not have any major comments on the manuscript but believe that the manuscript will benefit from incorporating the following suggestions:

- 1) Considering that the binding data for the Fab are more easily interpreted than the more complex data with the antibody, the authors might consider presenting the Fab characterization first. This would help to build confidence in the methods before the more complex data are considered.
- 2) In the extended data table 1, the legend reads "Kd (kinetics) was calculated from kon/koff" this should be changed to "Kd (kinetics) was calculated from koff/kon."
- 3) In the result section, the co-crystal structure of a J2 Fab bound to 23-bp dsRNA (line 177) says, "The structure captures two J2 Fabs bound to the same 23-bp dsRNA in crystallo". The electron density and the crystallographic model shows that the dsRNA bound to the Fabs is only 17/18 nt long. The result section should be modified to indicate that.
- 4) The method section for RNA preparation mentioned that the oligonucleotides were obtained from IDT and Dharmacon and were used without further purification. Did the authors verify the purity of the material? How might the purity of the material impact the BLI and FP experiments?
- 5) The BLI and CD experiments were performed at 2mM Mg⁺⁺ but other experiments were performed at 1mM Mg⁺⁺ concentration. Is there a significant impact on the helical nature of the nucleotides because of the difference in the amount of Mg⁺⁺ ions in solution?
- 6) In the extended data Table 1, please correct the number of significant figures for mutant HC Y50F.
- 7) According to the extended data table the kon and koff for WT 30mer dsRNA have substantial errors associated with the measurements. If possible, a more reliable measurement of kinetic parameters would be suitable.
- 8) Considering this paper is focused on recognition of RNA by Fab/antibody, the authors should be made aware of seven known RNA-Fab/antibody interfaces involving six different Fabs. Proc Natl Acad Sci U S A. 2008;105(1):82-7; Nat Struct Mol Biol. 2011;18(1):100-6; J Mol Biol. 2016;428(20):4100-14; Nat Commun. 2019;10(1):3629; ACS Chem Biol. 2020;15(1):205-16; Nat Chem Biol. 2022;18(4):376-84.

Reviewer #2

(Remarks to the Author)

I co-reviewed this manuscript with one of the reviewers who provided the listed reports. This is part of the Nature

Communications initiative to facilitate training in peer review and to provide appropriate recognition for Early Career Researchers who co-review manuscripts.

Reviewer #3

(Remarks to the Author)

This manuscript presents biophysical and structural analyses of the J2 antibody, a widely used tool for detecting dsRNA. The structure offers insights into the molecular basis of dsRNA recognition—clarifying how J2 distinguishes dsRNA from dsDNA, DNA:RNA hybrids and ssRNA. Given concerns about J2's specificity in complex samples, the structural data are valuable for interpreting potential cross-reactivity and guiding future antibody optimization.

While the work fills a knowledge gap, its impact is modest. The authors could strengthen the study by using the structural information to propose or test improved J2 variants with enhanced affinity or specificity.

Below are several points to address:

1. Potential for antibody optimization:

The authors report that J2 exhibits moderate affinity (~32nM for VA-I RNA and ~17nM for 50-bp dsRNA), which may limit its sensitivity in detecting endogenous dsRNA under pathological conditions. Researchers would benefit from insights into how J2 could be engineered for improved binding affinity.

The authors can analyze the dsRNA–J2–Fab interface more thoroughly and propose mutations that may stabilize the interaction. For example, in Fig. 4d, S33R nearly abolishes RNA binding, presumably due to steric clash. A substitution like S33K may retain a positive charge but with reduced bulk. Similarly, Y38 and Y101 likely participate in π – π or hydrophobic interactions. A Y101W and/or Y38W mutation could potentially strengthen stacking interactions and improve affinity.

The manuscript would be more compelling if it included rational mutagenesis strategies, guided by the structure, to create J2 variants with improved affinity to significantly enhance the impact of the work for tool and therapeutic development.

2. Affinity for poly(I:C)

Former studies mentioned J2 binds polyIC with approximately 10-fold lower affinity than it binds other dsRNAs. Since polyIC is widely used as a dsRNA mimic in both experimental and clinical settings (e.g., adjuvants), this observation needs further discussion. Can the authors elaborate on the structural or sequence-based reasons for this lower affinity? Addressing this would help researchers interpret J2 binding data in the context of poly(I:C) stimulation experiments.

3. A30P mutation and 3₁₀ Helix Integrity:

The A30P mutation disrupts the short 3₁₀ helix, but it also increases local hydrophobicity. The authors suggest this might compensate for the loss of secondary structure. To dissect these effects more clearly, they could test an A30G variant, which lacks the added hydrophobicity and destabilizes helical structures as well. This would better determine whether the helix is structurally required for RNA binding.

4. Stoichiometry of J2 Fab vs. IgG-dsRNA complex:

Depending on the assay, the reported binding stoichiometries vary between 2:1 and 1:1, which is confusing. Notably, the stoichiometry of IgG:dsRNA binding does not necessarily correspond to half that of Fab:dsRNA, even though each IgG has 2 arms of Fab. Based on the conformation of Fab:dsRNA interaction, can an IgG molecule simultaneously engage both of its Fab arms with the same dsRNA molecule? Alternatively, does it predominantly crossbridge two separate dsRNA molecules, effectively resulting in a 2:2 stoichiometry? Such crosslinking could lead to the formation of higher-order assemblies, and dsRNA length-dependent aggregate formation (as longer dsRNA would have higher propensity to form multi-dsRNA bridged structures), which would significantly complicate interpretation of the binding stoichiometry based on fluorescence polarization (FP) measurements.

Can the authors clarify these issues? Specifically, do they believe crossbridging has contributed to the reported stoichiometries and the apparent length selectivity of J2 binding? A direct comparison of dsRNA length sensitivity between J2 IgG and Fab would help address this point.

5. Comparison with other dsRNA antibodies

One additional aspect I would encourage the authors to consider is the utility of AlphaFold 3. Has the AlphaFold 3 -predicted structure of the J2-dsRNA complex been compared to the crystallographic structure? If so, how well do they agree? If AlphaFold 3 indeed reproduces the observed structure well, can the authors use structural prediction to model other dsRNA antibodies, such as J5, K1, or 9D5, which exhibit distinct sequence preferences (as noted in the manuscript, e.g., J2 and J5 prefer mixed base compositions, while K1 exclusively binds poly(I:C)).

Could structural modeling provide a rationale for these observed differences in sequence preference? For example, do differences in paratope orientation, groove access, or charge distribution explain K1's selective recognition of poly(I:C)? Furthermore, if AF3 or mutational modeling can reveal structure-function relationships across these antibodies, it may even be possible to rationally engineer a universal dsRNA antibody with improved tolerance to GC skew or secondary structure distortion and overcoming limitations such as those noted in the manuscript (e.g., "J2 exhibits substantial sensitivity to nucleotide composition and skew"). This could significantly enhance the manuscript's impact beyond structural characterization.

Reviewer #4

(Remarks to the Author)

The J2 antibody, which is derived from a mouse (IgG2a isotype), is commonly used in biological research to map and quantify double-stranded RNA (dsRNA), including in studies of virology, innate immune responses, and RNA biology. It is used in various applications, including ELISA, immunofluorescence, immunoprecipitation, dot blotting, and immunohistochemistry, and it is effective in both cultured cells and fixed tissue samples. J2 detects endogenous dsRNA, as well as dsRNA from viral intermediates, such as the hepatitis C and dengue viruses. Due to its widespread use, a detailed characterization of what the antibody detects and how it does so is of general interest.

The authors begin with a detailed biochemical characterization using biolayer interferometry. The data are compatible with a simple K_d -based binding model. The data confirm the field's consensus that J2 is highly specific to dsRNA. They also show that IC_{50} values decrease (affinity increases) as dsRNA length increases. While this result is likely what most biologists working with the antibody would have expected, it is unclear how it is to be reconciled with the footprint of J2 on the dsRNA, which corresponds to a much shorter region (see Figures 2 and 5).

The authors then proceed to a structural characterization (Figures 2 and 3). Towards the end of the work, the authors explore the dependence of binding on GC-content of the dsRNA, and they compare antibody strategies with the strategies of other (mostly immune related) proteins for dsRNA recognition. The authors argue that these strategies are different, and provide a rationale for the adoption of different strategies in terms of the relevant ionic milieu.

Questions and comments on merit:

How can the data on the dependence of IC_{50} values on the length of dsRNA be reconciled with the J2 footprint on dsRNA? Is the length dependence merely a result of longer dsRNA being able to bind to more than one copy of J2? In BLI experiments, a 2:1 stoichiometry is used for all analyses, irrespective of dsRNA length. Assuming this, it appears that the IC_{50} decreases, i.e., affinity increases, with increasing dsRNA length. However, the authors report counterexamples to the 2:1 model for the Adenovirus Associated RNA-I. I suspect the authors are on the wrong track altogether with the concept of dsRNA-length-independent stoichiometry to explain binding.

Instead, they should consider whether apparent increased affinity is simply a result of more J2 molecules binding to longer dsRNA while the affinity of each J2 molecule remains constant (provided there is enough dsRNA for the footprint). As far as I can tell, this model could explain the plateauing of the IC_{50} above 40 bp dsRNA because constant increases in dsRNA length increase the number of available binding sites by smaller and smaller multiplicative factors. Would this model be sufficient to explain the apparent increase in affinity?

If the "constant affinity per J2/length-dependent number of J2 binding sites on dsRNA" model cannot explain the apparent changes in IC_{50} , could the apparent increase in affinity be a consequence of favorable interactions between two or more J2 molecules bound to one longer dsRNA target? This could be tested by comparing a target with alternating dsRNA/dsDNA regions (so that bound J2 molecules cannot interact) with a target with blocked regions (so that J2 molecules can interact).

dsRNA cannot adopt a backbone conformation like "wet" (B-form) DNA, but conversely, DNA can adopt a dry (A-form) structure that has similar backbone parameters to dsRNA. What hinders it from doing so? I doubt that the free energy cost for the dsDNA B- to A-form conversion is high enough that it could be explain the affinity difference. This suggests that 2'-OH groups of the dsRNA are also directly recognized. Is this the case, and how much -based on an analysis of the structure- are 2'-OH group interactions expected to contribute to the affinity difference of J2 binding to dsRNA and dsDNA?

I like the argument that different ionic milieus call for different strategies of dsRNA recognition, due to enhancing effect of salt on hydrophobic interaction, and the screening effect on charge-charge interactions. Unfortunately, the authors are vague about the exact conditions. Can it be reconstructed from the way J2 made in the first place what ionic conditions J2 was "naturally" optimized for? This would make the comparison with the intracellular conditions more concrete, and the conceptually nice idea more convincing.

Presentation:

It has become customary to sweep the crystallographic details of work as in the present study completely under the table. I am not advocating for a full crystallographic table in the main text as in the past. But for manuscripts that are mainly about structure, a minimal crystallographic table with the key information (what space group, how many molecules in the asymmetric unit, what resolution, refined to what R_{free}) should be present in the main text for readers to have a rough idea of the quality of the structure. Also, I was unable to find a full crystallographic table in the Suppl. and could only assess the quality of the structure based on the PDB validation report. A full crystallographic table should still be included in the Suppl. To the authors credit, the data in the table in the validation report are good (for the experimental resolution), there is no need for additional structure refinement.

Version 1:

Reviewer comments:

Reviewer #1

(Remarks to the Author)

the authors have done a thorough job addressing my and my co-reviewer's comments. I am satisfied with the quality of this manuscript.

Reviewer #2

(Remarks to the Author)

I co-reviewed this manuscript with one of the reviewers who provided the listed reports. This is part of the Nature

Communications initiative to facilitate training in peer review and to provide appropriate recognition for Early Career Researchers who co-review manuscripts.

Reviewer #3

(Remarks to the Author)

The authors have adequately addressed all of my comments and suggestions.

Reviewer #4

(Remarks to the Author)

In the previous revision round, I was mainly concerned about

- the claim of 2:1 binding stoichiometry
- the apparent discrepancy between the antibody footprint and the length-dependence of the binding constant.

The first issue turned out to be a question of nomenclature, the 2:1 was meant to describe a non-homogeneous binding model. As the other referees were equally confused, it is good that the nomenclature has been changed. I have no objection to a non-homogeneous binding model.

The second issue has been very substantively addressed. The authors suggest a different explanation than the one that I had had in mind, and back it up with CD analysis of dsRNA of different lengths. While not entirely conclusive, their model is certainly reasonable.

With this, my reservations are fully addressed.

Point-by-point responses to reviewer comments:

Reviewer #1 (Remarks to the Author):

The manuscript by C. Bou Nader et al. provides a comprehensive and thorough analysis of the J2 monoclonal antibody which is used to detect endogenous and exogenous dsRNA. The study established J2 as a double stranded RNA (dsRNA) specific antibody that discriminates against other single stranded nucleotides and nucleotides of mixed composition. The mechanism of epitope binding by J2 antibody was revealed through binding experiments with various lengths of dsRNA and the co-crystal structure of the J2 antibody fragment with a dsRNA. The study expands the utility of J2 antibody as a dsRNA detection tool by determining the minimum required length of dsRNA for antibody binding while also highlighting the limitations of the tool by showing its strong preference for low GC content dsRNA. The study increases the reliability of dsRNA detection by J2 antibody.

The co-reviewers do not have any major comments on the manuscript but believe that the manuscript will benefit from incorporating the following suggestions:

A: We thank the reviewer for their positive assessments and helpful suggestions.

1) Considering that the binding data for the Fab are more easily interpreted than the more complex data with the antibody, the authors might consider presenting the Fab characterization first. This would help to build confidence in the methods before the more complex data are considered.

A: We appreciate the suggestion. Indeed, the characterizations started with the Fab (Fig. 1b and S1). We then intended to focus more on the IgG over the Fab, as the IgG form is what the research community nearly exclusively use, instead of the Fab. To reduce the apparent complexity and improve the clarity of the IgG BLI data, we have reworked the descriptions in the text, figures, and methods. The point of confusion stems from the misnomer of “2:1 model” used by Gator Bio to describe 1:1 binding to heterogeneous immobilized ligands (J2). However, “2:1 model” is almost always intuitively interpreted to mean stoichiometry, which is not the case here. We have substantially clarified this point of confusion throughout the manuscript, and eliminated the use of “2:1 model” and only refer to it as the “heterogeneous ligand” model. The “1:1” model is renamed “homogeneous ligand” model. Through direct consultations and a virtual meeting with Gator Bio technical team we have communicated these software and renaming suggestions to the manufacturer.

2) In the extended data table 1, the legend reads “Kd (kinetics) was calculated from kon/koff” this should be changed to “Kd (kinetics) was calculated from koff/kon.”

A: We thank the reviewer for catching this error, which has been corrected.

3) In the result section, the co-crystal structure of a J2 Fab bound to 23-bp dsRNA (line 177) says, “The structure captures two J2 Fabs bound to the same 23-bp dsRNA in crystallo”. The electron density and the crystallographic model shows that the dsRNA bound to the Fabs is only 17/18 nt long. The result section should be modified to indicate that.

A: We corrected and clarified this as suggested. Although the 23-bp dsRNA was used in the experiment, the terminal regions only have weak densities suggesting conformational flexibility. We now state:

“The structure captured two J2 Fabs bound to the opposite sides of a central, 17-bp portion of the same dsRNA in crystallo, via nearly identical interfaces (all-atom RMSD ~ 0.6 Å, Fig. 2a, b, d, Supplementary Fig. 5-6). While the central 17-bp dsRNA segment exhibited well-defined electron density, both flanking 3-bp terminal regions had only weak or no density. This finding is consistent with the notion that terminal regions of dsRNAs such as HIV-1 TAR exhibit substantial conformational flexibility and engage in transient excursions^{26,27}.”

4) The method section for RNA preparation mentioned that the oligonucleotides were obtained from IDT and Dharmacon and were used without further purification. Did the authors verify the purity of the material? How might the purity of the material impact the BLI and FP experiments?

A: The reviewer raised a valid consideration for oligo purity and quality. We have inspected all the HPLC mass spec analyses performed and provided by the manufacturers, which suggest a high degree of chemical purity and agreement with the expected molecular masses. Please see examples below, including the longest 50-nt oligos.

5) The BLI and CD experiments were performed at 2mM Mg⁺⁺ but other experiments were performed at 1mM Mg⁺⁺ concentration. Is there a significant impact on the helical nature of the nucleotides because of the difference in the amount of Mg⁺⁺ ions in solution?

A: Previous NMR and MD analyses suggested that Mg²⁺ ions strengthen helical stacking by neutralizing the negative charges of the phosphate backbone, which could impact the helical structure of various dsRNAs (e.g. PMIDs: 17325014, 12798678, 29718375). To address this question more explicitly, we performed additional temperature-scanning CD analyses of three representative dsRNA of low (0%), mid (47%), and high (100%) GC contents, at 0, 1, and 2 mM Mg²⁺. In all cases the CD spectra in 1 and 2 mM Mg²⁺ are nearly identical (Figure below), suggesting that within this range of Mg²⁺ concentration, the helical structure of the dsRNA under study remains similar, thus permitting comparison and correlation of data collected at 1 mM or 2mM Mg²⁺. By contrast, the CD spectra in the absence of Mg²⁺ show detectable, but relatively minor differences. This Mg²⁺ effect is more pronounced for low- and mid-GC% dsRNAs than the high-GC% dsRNA. This is likely due to the high-GC dsRNA already exhibiting stable A-form duplex structures without Mg²⁺. We further confirmed that where measurable, increasing Mg²⁺ concentration increased the T_m of the dsRNAs, consistent with known effects of Mg²⁺ on dsRNA duplex stability. These new data have been incorporated into a new Supplementary Fig. 14 that addresses Mg²⁺ effects on dsRNA geometry. We now state in a new main text section:

“Finally, we examined how Mg²⁺ may modulate the dsRNA helical structure⁴⁶⁻⁴⁸ and indirectly affect J2 interactions. We collected temperature-scanning CD spectra of 30-bp dsRNAs of low (0%), mid (47%), and high (100%) GC contents, at 0, 1, and 2 mM Mg²⁺. As expected, Mg²⁺ stabilized the A-form helical geometry of low- and mid-GC dsRNAs and raised their T_{ms}, while exerting minimal effects on the high-GC dsRNA that is already stable without Mg²⁺ (Supplementary Fig. 14). There was little difference in the CD spectra collected at 1 or 2 mM Mg²⁺. Together, these data suggest that RNA topology (e.g. poly(rI)-poly(rC)), dsRNA length, and solute conditions (including monovalent and divalent cations such as Mg²⁺) likely additionally impact J2 interactions.”

[Supplementary Fig. 14]

6) In the extended data Table 1, please correct the number of significant figures for mutant HC Y50F.

A: Corrected as suggested.

7) According to the extended data table the k_{on} and k_{off} for WT 30mer dsRNA have substantial errors associated with the measurements. If possible, a more reliable measurement of kinetic parameters would be suitable.

A: As suggested, we have repeated these measurements, which have substantially reduced the measurement errors and uncertainties.

8) Considering this paper is focused on recognition of RNA by Fab/antibody, the authors should be made aware of seven known RNA-Fab/antibody interfaces involving six different Fabs. Proc Natl Acad Sci U S A. 2008;105(1):82-7; Nat Struct Mol Biol. 2011;18(1):100-6; J Mol Biol.

2016;428(20):4100-14; Nat Commun. 2019;10(1):3629; ACS Chem Biol. 2020;15(1):205-16; Nat Chem Biol. 2022;18(4):376-84.

A: Added all these relevant references as suggested and new text near the end of the Results section. We now state:

“For comparison, multiple synthetic antibodies against structured RNAs discovered through phage display also extensively leverage non-polar interactions in addition to polar contacts³⁹⁻⁴⁵”

Reviewer #2 (Remarks to the Author):

A: We thank the reviewer for supporting the early career researchers.

Reviewer #3 (Remarks to the Author):

This manuscript presents biophysical and structural analyses of the J2 antibody, a widely used tool for detecting dsRNA. The structure offers insights into the molecular basis of dsRNA recognition—clarifying how J2 distinguishes dsRNA from dsDNA, DNA:RNA hybrids and ssRNA. Given concerns about J2’s specificity in complex samples, the structural data are valuable for interpreting potential cross-reactivity and guiding future antibody optimization. While the work fills a knowledge gap, its impact is modest. The authors could strengthen the study by using the structural information to propose or test improved J2 variants with enhanced affinity or specificity.

A: We thank the reviewer for their overall favorable assessments and helpful suggestions.

Below are several points to address:

1. Potential for antibody optimization:

The authors report that J2 exhibits moderate affinity (~32nM for VA-I RNA and ~17nM for 50-bp dsRNA), which may limit its sensitivity in detecting endogenous dsRNA under pathological conditions. Researchers would benefit from insights into how J2 could be engineered for improved binding affinity.

The authors can analyze the dsRNA–J2–Fab interface more thoroughly and propose mutations that may stabilize the interaction. For example, in Fig. 4d, S33R nearly abolishes RNA binding, presumably due to steric clash. A substitution like S33K may retain a positive charge but with reduced bulk. Similarly, Y38 and Y101 likely participate in π – π or hydrophobic interactions. A Y101W and/or Y38W mutation could potentially strengthen stacking interactions and improve affinity.

The manuscript would be more compelling if it included rational mutagenesis strategies, guided by the structure, to create J2 variants with improved affinity to significantly enhance the impact of the work for tool and therapeutic development.

A: We appreciate the insightful and specific suggestions from the reviewer. Indeed our structure and results have indicated potential pathways toward improving it. However, we believe that systemic optimization and improvement of the J2 antibody warrants a dedicated future study, and is beyond the intent and scope of this first manuscript. Our limited scope has been focused on quantitatively defining J2's nucleic acid-binding properties, revealing its biases and limitations, and elucidating its basic mechanisms of recognition, all of which will lay the groundwork for effective improvements in the future.

Second, regarding the proposed new mutants, given the presumed steric clash from the flexible side chain of Arg and considering its numerous rotamers, it is unclear to us that a Lys at S33 would necessarily fare much better. For the suggested Y101W and Y38W substitutions we were concerned that the substantial bulk of their side chains may disrupt the intimate, form-fitting interfaces with the minor groove and outweigh any potential gain in increased stacking strength.

Third, in our previous S9.6 antibody study (Bou-Nader 2022 *Nat Commun*), we did attempt to improve the antibody by more extensive rational design, which did not yield any improvements. In that study we substituted Y54 which makes a key sugar- π interaction responsible for selectivity of hybrids over dsRNA, individually with 11 other side chains. Unfortunately, none of the variants exhibited improved hybrid/dsRNA selectivity, including Y54F, Y54W, Y54H, Y54R, etc. Despite being imperfect, Tyr is already the best residue at this location, in its specific sequence context. Mirroring that study, our limited trials of structure-guided rational design here with S33R and N101R in J2 also failed to improve the antibody. The lesson we have gathered from both studies is that simple, structure-guided engineering of single side chains in antibody CDRs seems to have a low probability of success, probably due to the extremely limited sampling of the vast sequence space. By comparison, the initial creation of these antibodies through V(D)J recombination and somatic hypermutation had already sampled $\sim 5 \times 10^3 - 7 \times 10^4$ CDR variants. This might partially explain the challenges we have encountered in trying to improve these antibodies through simple point mutations. To achieve meaningful improvements that outperform the naturally selected antibodies, we believe that multiple residues and the length of the CDR loops need be systematically mutated and varied, screened, and selected towards the desired traits. AI tools that can rapidly sample large sequence space such as AlphaFold 3, in combination with our experimental complex structure, may assist in a future virtual screening study.

2. Affinity for poly(I:C)

Former studies mentioned J2 binds polyIC with approximately 10-fold lower affinity than it binds other dsRNAs. Since polyIC is widely used as a dsRNA mimic in both experimental and clinical settings (e.g., adjuvants), this observation needs further discussion. Can the authors elaborate on the structural or sequence-based reasons for this lower affinity? Addressing this would help researchers interpret J2 binding data in the context of poly(I:C) stimulation experiments.

A: Indeed, we agree with the reviewer that given its widespread use, it would be informative to test and understand J2 interactions with polyIC, despite it being an artificial dsRNA mimic. We have now performed CD and BLI analyses of polyIC binding to J2. Our CD spectra of polyIC is

nearly identical to previously published CD data (PMID: 25184857, 30079840), showing that polyIC only partially resembles natural dsRNA helices (please compare to the AU-only and GC-only dsRNA spectra in the Figure below). In particular, polyIC exhibits prominent 245 and 280 nm CD peaks, suggesting distinct exciton coupling from the stacked inosine-cytosine pairs. In particular, the unique peak at ~280 nm (and the absence of the 260 nm peak) indicates that free polyIC exhibits an unusual helical geometry, as is reminiscent of B-form dsDNA helices. This distinct geometry of polyIC is likely at least partially responsible for the reduced affinity in previous reports. However, accurate measurements of binding affinities to PolyIC is difficult, due to its variable length from 1.5 to 8 kb and structural heterogeneity. Due to difficulty in annealing long, repetitive sequences of polyI and polyC, polyIC is known to contain mixed dsRNA and ssRNA segments due to extensive pairing register shifts and loop extrusion. To confirm this we measured polyIC's melting temperature, which was 55.7 °C in our standard condition (in 25 mM Tris-HCl pH 7.5, 25 mM NaCl, and 2 mM EDTA), consistent with previous reports (e.g., PMID: 15600340). For comparison, perfectly paired dsRNAs between 20-30 bp can reach similar T_m s. The low T_m is consistent with the notion that polyIC does not contain very long contiguous dsRNA segments (e.g. >100 bp). Nonetheless, we tested polyIC binding to J2 using BLI, and found that polyIC does not visibly dissociate from immobilized J2. Its K_d could not be accurately determined but appears to be in pM range, lower than regular dsRNAs. This is likely due to crossbridging and avidity effects, where the multiple covalently tethered, relatively short dsRNA segments present in the same polyIC molecule bound multiple J2 IgGs on the sensor surface. We conclude that J2 is quite competent in detecting polyIC, but its precise binding affinity is uncertain and likely variable, due to polyIC's variable lengths, heterogeneous topological structure, unusual helical geometry, mixed dsRNA/ssRNA composition, and crossbridging effects. Notably, in a practical immunofluorescence-type of imaging experiment, where non-immobilized J2 IgG is added to cells transfected with polyIC, there would not be crossbridging. Under these conditions J2 IgG may exhibit reduced binding to polyIC due to its mixed dsRNA and ssRNA topology and geometric differences from natural dsRNAs evidenced by their unique CD spectra. However, it remains difficult to carry out a fair comparison with bona fide dsRNAs due to the numerous aforementioned differences. We have added a new text section and new Supplementary Fig. 12 (below) describing these characterizations of PolyIC-J2 interactions.

[Supplementary Fig. 12]

3. A30P mutation and 3₁₀ Helix Integrity:

The A30P mutation disrupts the short 3₁₀ helix, but it also increases local hydrophobicity. The authors suggest this might compensate for the loss of secondary structure. To dissect these effects more clearly, they could test an A30G variant, which lacks the added hydrophobicity and destabilizes helical structures as well. This would better determine whether the helix is structurally required for RNA binding.

A: We appreciate this thoughtful suggestion. Prolines in short helices such as 3₁₀ Helix would have almost certainly removed the helical structure. Given that A30P maintained dsRNA binding, it seems clear that the 3₁₀ Helix is not structurally required for RNA binding. A30G would not destabilize the 3₁₀ Helix as severely as A30P, nor add hydrophobicity due to its lack of side chain. We don't anticipate A30G would exert a strong effect either given the lack of effect of A30P, a result that may not be very helpful in further characterizing the interface.

4. Stoichiometry of J2 Fab vs. IgG-dsRNA complex:

Depending on the assay, the reported binding stoichiometries vary between 2:1 and 1:1, which is confusing. Notably, the stoichiometry of IgG:dsRNA binding does not necessarily correspond to half that of Fab:dsRNA, even though each IgG has 2 arms of Fab. Based on the conformation of Fab:dsRNA interaction, can an IgG molecule simultaneously engage both of its Fab arms with the same dsRNA molecule? Alternatively, does it predominantly crossbridge two separate dsRNA molecules, effectively resulting in a 2:2 stoichiometry? Such crosslinking could lead to the formation of higher-order assemblies, and dsRNA length-dependent aggregate formation (as

longer dsRNA would have higher propensity to form multi-dsRNA bridged structures), which would significantly complicate interpretation of the binding stoichiometry based on fluorescence polarization (FP) measurements. Can the authors clarify these issues? Specifically, do they believe crossbridging has contributed to the reported stoichiometries and the apparent length selectivity of J2 binding? A direct comparison of dsRNA length sensitivity between J2 IgG and Fab would help address this point.

A: Indeed, this was an anticipated major point of confusion, which we tried inadequately to preempt. There was no 2:1 stoichiometry in this manuscript. As mentioned in responses to Reviewer #1 point 1, the 2:1 (referring to BLI binding model) is an unfortunate misnomer used by Gator Bio to describe two types of ligands (heterogeneous ligand) each binding 1:1 to an incoming analyte, or the presence of two distinct binding modes. We had tried to explain and clarify this unusual use of 2:1 in the Methods but confusion was inevitable, as 2:1 nearly always refers to stoichiometry. We have thus removed all mentions of 2:1, and only describe this as “heterogeneous ligand” model.

As to whether the two Fab arms of the same J2 IgG can bind to the same dsRNA (≤ 50 bp) in our experiments, we believe the answer is no, except for perhaps the newly added polyIC experiment. The two Fab arms of IgG are flexibly tethered to the central constant region via hinges. However, the substantial bulk and rigidity of each Fab arm creates substantial steric hindrance restricting the relative motion between the two Fab arms. It has been estimated using programmable DNA origami-tethered epitopes, synthetic antigen arrays or viral capsid surfaces, that bivalent IgG binding avidity is highest when the two epitopes were separated by 10-15 nm, which covers 36-58 bp (PMIDs: 24997862, 32561744, 30643273, 34135438). This separation, plus two 8-bp footprints of two Fabs, would mean that 52 -74 bp dsRNA would be minimally needed to engage both Fabs of the same IgG at the same time. In reality even longer dsRNA is probably required when there is no flexible ssRNA linker present between the two Fab-bound dsRNA segments, as the high rigidity of the dsRNA (persistent length ~ 250 bp) imposes certain orientations of the Fabs, further hindering IgG bivalency. In conclusion, in our experiments where relatively short dsRNAs are used (≤ 50 bp), bivalent J2 IgG binding to the same dsRNA is probably minimal. For polyIC, we observed no dissociation from J2 IgG, which is consistent with crossbridging between different Fab arms or IgGs. In addition, our AUC and SEC-MALS analyses, using dsRNAs 23-30bp in length, only observed 1:1 stoichiometry between dsRNA and J2 Fab/IgG (Supplementary Fig. 1). We have added clarifications, discussions, and references on this point to the text.

5. Comparison with other dsRNA antibodies

One additional aspect I would encourage the authors to consider is the utility of AlphaFold 3. Has the AlphaFold 3 -predicted structure of the J2-dsRNA complex been compared to the crystallographic structure? If so, how well do they agree? Could structural modeling provide a rationale for these observed differences in sequence preference? For example, do differences in paratope orientation, groove access, or charge distribution explain K1's selective recognition of poly(I:C)? Furthermore, if AF3 or mutational modeling can reveal structure-function relationships across these antibodies, it may even be possible to rationally engineer a universal dsRNA antibody with improved tolerance to GC skew or secondary structure distortion and

overcoming limitations such as those noted in the manuscript (e.g., “J2 exhibits substantial sensitivity to nucleotide composition and skew”). This could significantly enhance the manuscript’s impact beyond structural characterization.

A: We agree completely with the reviewer that AF3 modeling should now be routinely performed to compare with experimental structures to evaluate prediction success, give feedback to the prediction community, and to receive inspiration and alternative ideas. As suggested, we have now performed AF3 predictions for the J2 Fab-dsRNA interface. While AF3 did a remarkable job predicting the free J2 Fab structure (panel **a** below), including the CDRs (overall RMSDs ~ 0.9 - 1.0 Å), and formation of dsRNA duplex as expected, it is completely unable to predict the J2-dsRNA interface (panel **b**). This failure is probably attributable to insufficient training from the limited PDB data on antibody-nucleic acid structures. Notably AF3 is apparently unaware of the fact that CDR regions would be expected to interact with the dsRNA epitope. The stark contrast between AF3’s impressive success to predict free J2 Fab structure including the CDRs and complete failure in protein-dsRNA interfaces highlight the continued need for experimental determination of protein-nucleic acids structures, especially of the relatively scarce antibody-nucleic acid structures. We included these data in a new Supplementary Fig. 9 (below), and descriptions in the text.

[Supplementary Fig. 9]

Now we state:

“Finally, we evaluated how well AlphaFold 3 can predict the J2-dsRNA interface. Interestingly, AlphaFold3 successfully predicts the conformation of the J2 Fab including the CDRs, achieving

a RMSD of 0.9-1.0 Å. However, it completely fails to predict the J2-dsRNA interface, placing dsRNA in various orientations and locations far from the CDRs (Supplementary Fig. 9). This finding attests to the remarkable ability of AlphaFold 3 to predict protein structures and conformations including antibody CDRs but also highlights its lack of training in unconventional RNA-protein interfaces such as antibody-nucleic acids interfaces underrepresented in the PDB.”

If AlphaFold 3 indeed reproduces the observed structure well, can the authors use structural prediction to model other dsRNA antibodies, such as J5, K1, or 9D5, which exhibit distinct sequence preferences (as noted in the manuscript, e.g., J2 and J5 prefer mixed base compositions, while K1 exclusively binds poly(I:C)).

A: As shown above AF3 completely fails to pinpoint the interface, or even come up with some plausible models with J2-dsRNA proximity. As far as we know, there is no amino acid sequence available for any other dsRNA antibodies such as J5, K1, or 9D5. We have already shared J2 plasmids and sequences with several research groups. In future studies, we do aim to continue to sequence and investigate additional nucleic acid antibodies and make them freely available to the research community.

Reviewer #4 (Remarks to the Author):

The J2 antibody, which is derived from a mouse (IgG2a isotype), is commonly used in biological research to map and quantify double-stranded RNA (dsRNA), including in studies of virology, innate immune responses, and RNA biology. It is used in various applications, including ELISA, immunofluorescence, immunoprecipitation, dot blotting, and immunohistochemistry, and it is effective in both cultured cells and fixed tissue samples. J2 detects endogenous dsRNA, as well as dsRNA from viral intermediates, such as the hepatitis C and dengue viruses. Due to its widespread use, a detailed characterization of what the antibody detects and how it does so is of general interest.

The authors begin with a detailed biochemical characterization using biolayer interferometry. The data are compatible with a simple K_d-based binding model. The data confirm the field's consensus that J2 is highly specific to dsRNA. They also show that IC₅₀ values decrease (affinity increases) as dsRNA length increases. While this result is likely what most biologists working with the antibody would have expected, it is unclear how it is to be reconciled with the footprint of J2 on the dsRNA, which corresponds to a much shorter region (see Figures 2 and 5).

The authors then proceed to a structural characterization (Figures 2 and 3). Towards the end of the work, the authors explore the dependence of binding on GC-content of the dsRNA, and they compare antibody strategies with the strategies of other (mostly immune related) proteins for dsRNA recognition. The authors argue that these strategies are different, and provide a rationale for the adoption of different strategies in terms of the relevant ionic milieu.

A: We thank the reviewer for their detailed analyses, insightful syntheses, favorable assessments and helpful suggestions.

Questions and comments on merit:

How can the data on the dependence of IC₅₀ values on the length of dsRNA be reconciled with the J2 footprint on dsRNA? Is the length dependence merely a result of longer dsRNA being able to bind to more than one copy of J2? In BLI experiments, a 2:1 stoichiometry is used for all analyses, irrespective of dsRNA length. Assuming this, it appears that the IC₅₀ decreases, i.e., affinity increases, with increasing dsRNA length. However, the authors report counterexamples to the 2:1 model for the Adenovirus Associated RNA-I. I suspect the authors are on the wrong track altogether with the concept of dsRNA-length-independent stoichiometry to explain binding.

A: We do apologize for the across-the-board confusion about the intended meaning of “2:1”. The binding stoichiometry is always 1:1 in this manuscript except for possibly polyIC (new Supplementary Fig. 12). As discussed in the responses to Reviewer #1 point 1 and Reviewer #3 point 4, the “2:1” refers to a heterogeneous ligand model, not binding stoichiometry. We have now removed any such use of “2:1”, which is an unfortunate misnomer used by the instrument manufacturer’s software. Their intention was that there were two types of ligands (immobilized J2) present on the surface, or two distinct modes of binding — thus heterogeneous ligands (J2) receiving a homogeneous analyte (dsRNA). We completely agree that it should not be referred as “2:1”, and now only refer to it as “heterogeneous ligand” model.

The reviewer raised an important question about the observed dsRNA length dependency of J2 (~30 bp for near optimal binding) significantly exceeding the structural footprint (8 bp). The first, simplest explanation to consider would be stoichiometry. However, our AUC and SEC-MALS analyses, using dsRNAs 23-30 bp in length, only observed 1:1 stoichiometry between dsRNA and J2 Fab/IgG (Supplementary Fig. 1).

A second possible explanation would be dsRNA helical geometry. We reason that as dsRNA increases in length, the accumulative increase in overall stacking strength would cause the dsRNA to axially compress and shorten, which would in turn alter the width and geometry of the minor groove and impact J2 binding. We know by comparing apo and hybrid-bound S9.6 antibody structures that the CDRs are not very flexible and preconfigured for the epitope, which results from the process of affinity maturation. If J2 CDRs are preconfigured to bind a fixed minor groove geometry corresponding to its original immunogen, which are natural, 4.6-kb long dsRNAs, they might not fit potentially different helical geometry of short dsRNAs.

To test this hypothesis experimentally, we performed CD analyses of the dsRNAs ranging from 6 to 50 bp in length, normalizing their concentrations so that each sample contained the same number of total nucleotides. Consistent with the hypothesis, we found that short dsRNAs that bind J2 poorly exhibit distinct CD spectra from the long dsRNAs that bind J2 strongly. Short dsRNAs exhibited reduced A-form characteristics including the valley at 210 nm, peak near 260 nm, and increased B-form characteristics such as peaks near 275 nm. The latter is further consistent with reduced stacking strength and increased helical rise found in dsDNA (~3.4 Å compared to 2.9 Å of A-form dsRNA). We propose a simple explanation that short dsRNAs are inefficiently self-stacked and thus lack the optimal minor groove geometry found on long dsRNAs required for robust J2 binding. Further, dsRNA overall geometry may be additionally affected by terminal regions that are transiently frayed or otherwise not stably paired, as seen in

the J2-dsRNA co-crystal structure and the HIV-1 TAR lower stem (Bou-Nader et al., 2025, Nat. Commun.). We have incorporated these new data to form a new Supplementary Fig. S13, shown below, and describe them in a new text section entitled “Effects of dsRNA length and Mg²⁺ on RNA helical geometry”.

[Supplementary Fig. 13]

Instead, they should consider whether apparent increased affinity is simply a result of more J2 molecules binding to longer dsRNA while the affinity of each J2 molecule remains constant (provided there is enough dsRNA for the footprint). As far as I can tell, this model could explain the plateauing of the IC₅₀ above 40 bp dsRNA because constant increases in dsRNA length

increase the number of available binding sites by smaller and smaller multiplicative factors. Would this model be sufficient to explain the apparent increase in affinity?

A: Please refer to the response to the preceding question. Based on recent studies on IgG bivalency and Fab spacing (please see responses to Reviewer #3, point 4), we believe 40-50 bp dsRNA is still too short for bivalent IgG binding to the same dsRNA. As to whether two J2 Fabs can bind to opposite sides of dsRNA as seen in the crystals, we believe that is feasible. However, our AUC and SEC-MALS analyses using dsRNAs of 23, 27, and 30 bp did not provide evidence for multiple J2 Fab or IgG binding to the same dsRNA.

If the "constant affinity per J2/length-dependent number of J2 binding sites on dsRNA" model cannot explain the apparent changes in IC_{50} , could the apparent increase in affinity be a consequence of favorable interactions between two or more J2 molecules bound to one longer dsRNA target? This could be tested by comparing a target with alternating dsRNA/dsDNA regions (so that bound J2 molecules cannot interact) with a target with blocked regions (so that J2 molecules can interact).

A: Please refer to the responses to the two preceding questions. We appreciate the innovative experimental design. Based on CD analyses in our previous S9.6 antibody study (Bou-Nader 2022 *Nat Commun*), mixed DNA/RNA composition could also lead to an overall mixed A/B-form intermediate helical geometry which impacted S9.6 binding. Adjacent dsRNA and dsDNA segments may affect each other's helical geometry through coaxial stacking. Such an experiment may produce complex results that are difficult to interpret, due to the contiguous grooves with possible gradual geometric transitions rather than sharp dsDNA/dsRNA boundaries. J2-J2 interactions on dsRNA is certainly a possibility. We observed what appeared to be dsRNA-triggered J2 IgG aggregation in some early experiments. We speculate that dsRNA-binding may have exposed certain hydrophobic residues driving unscheduled protein-protein interactions.

dsRNA cannot adopt a backbone conformation like "wet" (B-form) DNA, but conversely, DNA can adopt a dry (A-form) structure that has similar backbone parameters to dsRNA. What hinders it from doing so? I doubt that the free energy cost for the dsDNA B- to A-form conversion is high enough that it could explain the affinity difference. This suggests that 2'-OH groups of the dsRNA are also directly recognized. Is this the case, and how much -based on an analysis of the structure- are 2'-OH group interactions expected to contribute to the affinity difference of J2 binding to dsRNA and dsDNA?

A: Indeed, 5 2'-OH groups are directly recognized by each J2 Fab (Fig. 2f), with 2 and 3 on each strand of dsRNA. Mutational analyses of the 2'-OH-binding J2 residues show variable importance and contributions (Fig. 3d). HC N101 touches two 2'-OHs and is among the most important residue. HC N55 binds one 2'-OH and is functionally important. By contrast, the remaining two 2'-OHs, bound by LC Y31 and HC Y50, make minimal contributions, as their removal by Phe substitutions (LC Y31F, HC Y50F) conferred no binding defects. We had thought about removing specific 2'-OH groups on the dsRNA side to evaluate their contributions, as we did in the S9.6-hybrid study (Bou-Nader 2022 *Nat. Commun.*) However, we realized that J2 could simply bind elsewhere if we introduce a small number of deoxy modifications. If we then remove larger number of 2'-OH groups then it is expected to drive the duplex towards B-

form, based on our observations in the S9.6 study.

I like the argument that different ionic milieus call for different strategies of dsRNA recognition, due to enhancing effect of salt on hydrophobic interaction, and the screening effect on charge-charge interactions. Unfortunately, the authors are vague about the exact conditions. Can it be reconstructed from the way J2 made in the first place what ionic conditions J2 was "naturally" optimized for? This would make the comparison with the intracellular conditions more concrete, and the conceptually nice idea more convincing.

A: We appreciate the reviewer's insight and helpful suggestion on this idea. The typical ionic conditions under which antibodies such as J2 was naturally selected and optimized are those in extracellular/interstitial fluids bathing the lymph node follicles. They are characterized by ~150 mM ionic strength ($\text{Na}^+ \sim 140 \text{ mM}$, $\text{K}^+ \sim 4 \text{ mM}$, $\text{Ca}^{2+} \sim 1.3 \text{ mM}$, $\text{Mg}^{2+} \sim 1 \text{ mM}$, $\text{Cl}^- \sim 110 \text{ mM}$), pH 7.4, 37 °C, and are mildly oxidizing and protein-rich. We have added more detailed and specific information to the discussion to support the ionic milieu idea. Now we state:

“While intracellular RNA-binding proteins function in the nucleoplasm and cytoplasm, antibodies first undergo affinity maturation in germinal centers inside lymph node follicles, and then function in blood plasma, lymph, and interstitial fluids. Both the germinal centers and extracellular fluids have higher ionic strengths than the cytoplasm (~150 mM versus ~100 mM), typically contain $\text{Na}^+ \sim 140 \text{ mM}$, $\text{K}^+ \sim 4 \text{ mM}$, $\text{Ca}^{2+} \sim 2 \text{ mM}$, $\text{Mg}^{2+} \sim 1 \text{ mM}$, $\text{Cl}^- \sim 110 \text{ mM}$, $\text{HCO}_3^- \sim 25 \text{ mM}$, and are mildly oxidizing⁵⁹. Importantly, they primarily contain small, fully dissociated ions (Na^+ , Cl^- , HCO_3^-) that contribute strongly to ionic strength, which effectively suppress nonspecific electrostatic interactions. This makes antibodies that leverage hydrophobic interactions more effective and specific for nucleic acid interactions. By contrast, intracellular fluids are K^+ -rich and Na^+ -poor, and contain mostly large organic anions (ATP, proteins, nucleic acids) that are not fully dissociated and less mobile. Such conditions favor dynamic electrostatic interactions between proteins (e.g. basic residues) and nucleic acids due to the reduced shielding of their charges.”

Presentation:

It has become customary to sweep the crystallographic details of work as in the present study completely under the table. I am not advocating for a full crystallographic table in the main text as in the past. But for manuscripts that are mainly about structure, a minimal crystallographic table with the key information (what space group, how many molecules in the asymmetric unit, what resolution, refined to what Rfree) should be present in the main text for readers to have a rough idea of the quality of the structure. Also, I was unable to find a full crystallographic table in the Suppl. and could only assess the quality of the structure based on the PDB validation report. A full crystallographic table should still be included in the Suppl. To the authors credit, the data in the table in the validation report are good (for the experimental resolution), there is no need for additional structure refinement.

A: We agree with the reviewer. Assuming we have the permission of the editor, we have moved the X-ray crystallographic table to the main text to become Table 1.